# Identification and establishment of type IV interferon and the characterization of interferon-υ including its class II cytokine receptors IFN-υR1 and IL-10R2

Shan Nan Chen[1,2,3], Zhen Gan[1,7], Jing Hou[1,7], Yue Cong Yang[1], Lin Huang[1], Bei Huang[1,6], Su Wang[4,5] & Pin Nie [1,2,3,4,5✉]

Interferons (IFNs) are critical soluble factors in the immune system and are composed of three types, (I, II and III) that utilize different receptor complexes IFN-αR1/IFN-αR2, IFN-γR1/IFN-γR2, and IFN-λR1/IL-10R2, respectively. Here we identify IFN-υ from the genomic sequences of vertebrates. The members of class II cytokine receptors, IFN-υR1 and IL-10R2, are identified as the receptor complex of IFN-υ, and are associated with IFN-υ stimulated gene expression and antiviral activity in zebrafish (*Danio rerio*) and African clawed frog (*Xenopus laevis*). IFN-υ and IFN-υR1 are separately located at unique and highly conserved loci, being distinct from all other three-type IFNs. IFN-υ and IFN-υR1 are phylogenetically clustered with class II cytokines and class II cytokine receptors, respectively. Therefore, the finding of this IFN ligand-receptor system may be considered as a type IV IFN, in addition to the currently recognized three types of IFNs in vertebrates.

[1] State Key Laboratory of Freshwater Ecology and Biotechnology, Institute of Hydrobiology, Chinese Academy of Sciences, Wuhan, Hubei 430072, China. [2] Key Laboratory of Aquaculture Disease Control, Institute of Hydrobiology, Chinese Academy of Sciences, Wuhan, Hubei 430072, China. [3] Innovation Academy of Seed Design, Chinese Academy of Sciences, Wuhan, China. [4] Laboratory for Marine Biology and Biotechnology, Pilot National Laboratory for Marine Science and Technology (Qingdao), Qingdao, Shandong 266237, China. [5] School of Marine Science and Engineering, Qingdao Agricultural University, Qingdao, Shandong 266109, China. [6] Present address: College of Fisheries, Jimei University, 43 Yindou Road, Xiamen, Fujian 361021, China. [7] These authors contributed equally: Zhen Gan, Jing Hou. ✉email: pinnie@qau.edu.cn

The alpha-helical cytokines are divided into two classes, i.e., I and II, based mainly on the utilization of their receptors[1,2], and in mammals, class II cytokines contain interleukins (ILs), including IL-10, IL-19, IL-20, IL-22, IL-24, IL-26, and interferons (IFNs), including type I, type II and type III IFNs, which all play important roles in immune system via binding to specific class II cytokine receptors to regulate the expression of immune-related molecules[2–4]. Fish are also reported to have class II cytokines and their receptors[5–7]. So far, it is reported that fish possess class II cytokines, including IL-10, IL-20-like (IL-20L), IL-22, IL-26, and types I, II and III IFNs[6–8]. But, class II cytokine receptor family in teleost fish is named uniquely as cytokine receptor family B (CRFB)[9], with seventeen members in comparison with the well-characterized twelve members in humans[10,11].

Type I IFNs are produced upon viral infection, leading to the modulation of IFN-stimulated genes (ISGs) via type I IFN receptors, IFNAR1 and IFNAR2[9]. In placental mammals, type I IFNs contains a few subtypes, including IFN-α, IFN-β, IFN-δ, IFN-ε, IFN-ζ, IFN-κ, IFN-μ, IFN-ν, IFN-τ, and IFN-ω[12,13]. Although some type I IFN subtypes have a species-specific distribution, such as IFN-μ in horse (Equus caballus) and IFN-τ in ruminants[13,14], all these subtypes are intronless genes, which are clustered together on a conserved locus in mammalian species with annotated type I IFN loci[15,16]. In fact, the intronless type I IFN locus, which is PTPLAD2 - type I IFNs - MTAP gene cluster, is highly conserved from green anole (Anolis carolinensis, a reptile species) to human[7,17,18], but not in tropical clawed frog (Xenopus tropicalis, an amphibian species) which has an intron-containing, five-exon and four-intron type I IFN locus with conserved colinearity when compared to zebrafish (Danio rerio), and has an intronless type I IFN locus without any synteny compared to other vertebrates[17,18]. A hypothesis to explain how introns might have been lost from type I IFNs in amniotes is that intronless type I IFNs might have originated from intron-containing type I IFN transcripts via evolutionary retroposition event in a transition period when vertebrates migrated from an aquatic environment to land[17–19]. In zebrafish, only four copies of IFNs, which were previously named as IFN-φ1, φ2, φ3 and φ4[20], also known as IFN1, 2, 3 and 4, have multi-exon gene organization as observed in clawed frog[17,18], and are confirmed to be type I IFNs in consideration of sequence feature, phylogeny and protein crystal structure[15,21,22]. Moreover in teleost fish, type I IFN receptors contain two molecules, CRFB1 and CRFB2, equivalent phylogenetically and functionally to IFNAR2 in mammals, which are also known as IFNφ1-R1/IFNAR2-1 and IFNφ2-R1/IFNAR2-2, and are likely derived from gene duplication, and a single CRFB5 also known as IFNφ-R2, equivalent to IFNAR1[10,15,20,23]. In zebrafish, the four type I IFNs can be further divided into two groups, group I and group II, with IFN-φ1/φ4 and φ2/φ3, respectively, which signal through the receptor complexes, CRFB1 + CRFB5 and CRFB2 + CRFB5, respectively, to produce antiviral activity[15,20].

The type II IFN has only a single member, IFN-γ in mammals, which is involved in a variety of biological processes through the receptors, IFN-γRs, IFNGR1 and IFNGR2, and JAK/STAT signal pathway[24–26]. Human IFN-γ gene is linked to IL22 and IL26 on human chromosome 12 and has highly conserved synteny with other vertebrates[24,27], such as chicken (Gallus gallus), green anole, clawed frog and zebrafish, although type II IFNs in teleost fish consist of two members, IFN-γ (also named as IFN-γ2) and IFN-γ related molecule (IFN-γrel, also called IFN-γ1)[6,28]. IFNGR1 and IFNGR2 genes are also conserved in vertebrates, but duplication event may have resulted in the generation of two IFNGR1 genes, IFNGR1-1 (also named as CRFB17 in fish) and IFNGR1-2 (CRFB13) in teleost[15,28]. In zebrafish, IFNGR1-1, IFNGR1-2 and

IFNGR2 (CRFB6) are necessary for IFN-γ to induce the expression of ISGs, and IFN-γrel needs IFNGR1-1[10]. Although zebrafish IFN-γ and IFN-γrel interact with different receptor complex, the two type II IFNs show similarly functional aspects in the regulation of ISG expression and in resistance to microbial infections, and IFN-γ is involved in the regulation of MHCII expression[29,30].

Human type III IFNs comprise four members, IFN-λ1 (also named IL-29), IFN-λ2 (IL-28A), IFN-λ3 (IL-28B) and IFN-λ4, with same receptor system, IFN-λR1 and IL-10R2[31,32]. Multi-exon type III IFNs and their receptor genes (IFN-λR1 and IL-10R2) have also been identified in tetrapod species, such as chicken, green anole and clawed frog[19,33,34]. To date, there are no reports on teleost fish type III IFNs, although IFNL and IFNLR1 have been identified in cartilaginous fish[8,35]. In zebrafish, CRFB14 locus is syntenic to IL22RA1-IFNLR1 locus in shark and human[35]; however, it is still ambiguous that zebrafish CRFB14 is the orthologue of IL22RA1 or IFNLR1.

In addition, the characterized class II cytokine receptors in teleost fish also include IL10R1 (CRFB7), IL10R2 (likely homologous to CRFB4), IL20R1 (CRFB8), IL20R2 (CRFB16), IL22BP (CRFB9), Tissue Factor a (TFa, CRFB10) and Tissue Factor b (TFb, CRFB11)[5,10,36–40]. However, some fish class II cytokine receptors cannot be classified into orthologues of the known class II cytokine receptors in mammals (Supplementary Table 1). It can be hypothesized that the uncharacterized class II cytokine receptors may interact with unknown cytokine(s) in teleost fish. Hence, the identification, function and evolution of a formerly unreported class II cytokine, which is designated as IFN-υ (ifnu), and its receptors, including IFN-υR1 (ifnur1, also known as crfb12 in fish) and CRFB4/IL-10R2, were characterized in zebrafish. The IFN-υ in zebrafish was found to be responsive to virus infection, and to regulate the expression of ISGs and to inhibit virus replication. The IFN-υ and IFN-υR1 were also identified bioinformatically in different lineages of vertebrates including fish, amphibian, reptile, avian and mammal. Additionally, African clawed frog IFN-υ was found also to use IFN-υR1 and IL-10R2 (homologous to CRFB4) to promote antiviral state. It is intriguing that this IFN cannot be classified into any known types of IFNs, not types I, II, nor type III IFNs based on sequence feature, genetic locus, phylogeny and receptor usage, and it should be considered as the fourth type of IFN in addition to the already known three types of IFNs. This study thus contributes to the understanding of IFN systems and their evolution in vertebrates.

## Results

**Sequence characteristics of IFN-υ.** Since some fish genes belonging to class II cytokines cannot be well annotated due to their low expression levels, bioinformatic strategies (see Methods) were employed to scan available genomic sequences, and unannotated sequences in intergenic regions between annotated genes in zebrafish genome (assembly version: Zv9) were collected to carry out gene prediction and/or annotation on the basis of class II cytokine gene features, including 1) five-coding-exon organization, 2) intron phase being zero, 3) signal peptide present at N-terminal region, and 4) putative protein sequence containing multiple alpha helices. As a result, an unannotated gene, designated as IFN-υ, ifnu, was identified in zebrafish genome (on Chr 24), and its full-length cDNA sequence was cloned by using RACE PCR and deposited in GenBank with the accession number, MW547062. The IFN-υ ORF contains 492 bp and is predicted to encode a 163 amino acid protein with a putative signal peptide at N-terminal region. Multiple sequence alignment revealed that IFN-υ shares low identity with class II family cytokines including type I IFNs (12.9–23.3%), IFN-γ (7.4%) and

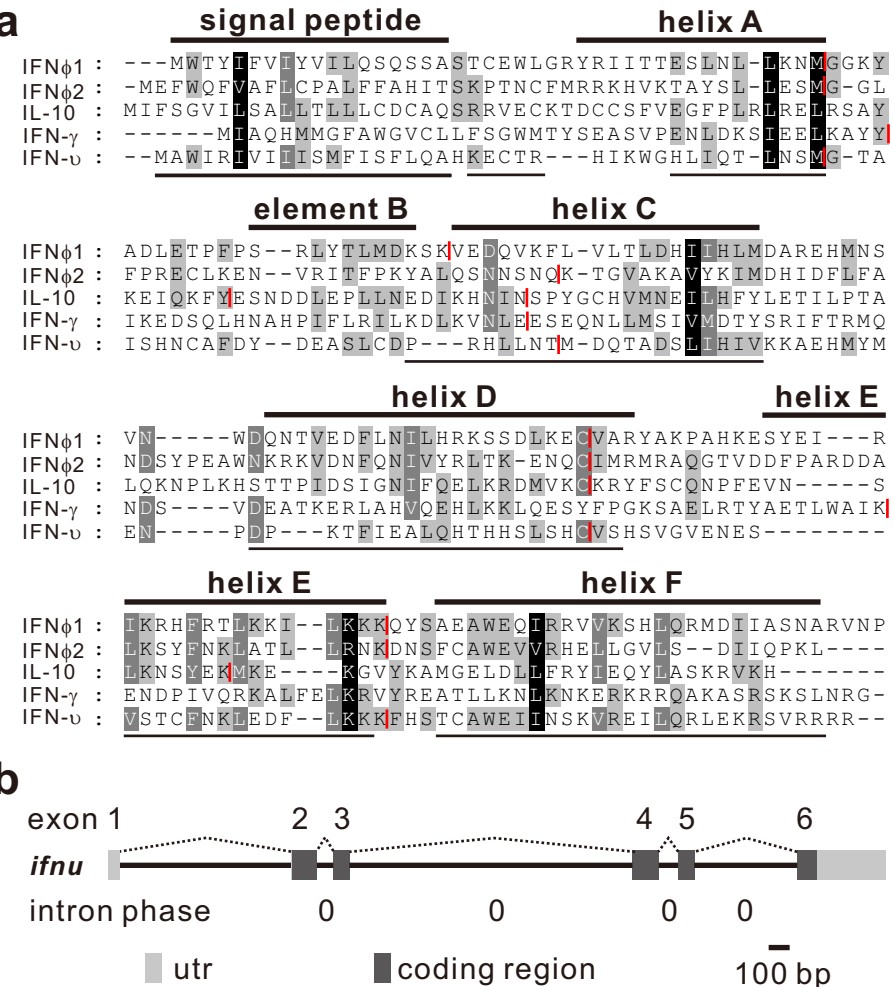

**Fig. 1 Identification of zebrafish IFN-υ. a** Sequence alignment of IFN-υ protein. Identical or similar amino acids are indicated in shade. The putative signal peptide and helices of IFN-υ are underlined and the known helices A to F for zebrafish IFNφ1 are highlighted by solid lines above the alignment. Intron positions are indicated in red line. **b** The gene organization of zebrafish IFN-υ. Exons and introns are indicated with gray boxes and lines, respectively. The intron phases of *ifnu* are all zero. The utr indicates untranslated regions.

IL-10 (4.8%) in zebrafish, although the C-terminal sequence of IFN-υ is similar to type I IFNs (Fig. 1a). IFN-υ was predicted to contain multi-helix protein structure (Fig. 1a) and consists of five-coding exons with all intron phases being zero (Fig. 1a, b), which is similar to the gene organization found in fish type I IFNs, IL-10 related cytokines and multi-exon type III IFNs in vertebrates, but not to the organization of the intronless type I IFNs in amniotes.

**Induction of ISGs and antiviral state in wildtype, over-expressed and *ifnu*⁻/⁻ zebrafish.** IFNs can induce antiviral state in vertebrates to suppress virus replication through the regulation of ISG expression. As shown in Fig. 2a, significantly increased expression of IFN-υ gene was observed at mRNA level in response to grass carp reovirus (GCRV) infection. Using RT-PCR, the increase in the expression of known fish anti-viral genes, was detected, including GCRV-induced gene 2 (*gig2*), interferon regulatory factor 1 (*irf1*), myxovirus (influenza) resistance A (*mxa*), and viperin (also known as radical S-adenosyl methionine domain containing 2, *rsad2*)[41–44], in zebrafish embryos with the overexpression of IFN-υ (Fig. 2b and Supplementary Fig. 1a–d). Simultaneously, the antiviral activity of IFN-υ was examined, and the titer of GCRV was significantly reduced in zebrafish larvae (Fig. 2c).

Furthermore, *ifnu* knockout zebrafish was generated to verify the function of IFN-υ. Cas9/gRNA system specific to *ifnu* was developed to target the gene Exon 4, which led to 17 bp deletion, and resulted in the mutation of frameshift and premature translation termination (Supplementary Fig. 2a). The expression of IFN-υ gene decreased significantly in *ifnu* knockout zebrafish *ifnu*⁻/⁻, in comparison with the wildtype (WT) after GCRV infection (Supplementary Fig. 2b). As expected, *ifnu*⁻/⁻ zebrafish showed increased susceptibility to GCRV infection. IFN-υ deficiency resulted in significantly down-regulated expression of anti-viral ISGs, including *gig2*, *irf1*, *mxa* and *viperin*, in response to GCRV infection (Fig. 2d−g), and also significantly increased GCRV titer when compared with the WT (Fig. 2h). In addition, the knockdown of type I (IFN-φ1) and type II IFN (IFN-γ and IFN-γrel) in *ifnu*⁻/⁻ zebrafish further increased the susceptibility to GCRV infection (Supplementary Fig. 3), suggesting that the antiviral signaling of these IFNs can be relatively different.

**IFN-υ receptors, CRFB12 (IFN-υR1) and CRFB4 (IL-10R2).** Specific class II cytokine receptors are essential for the functioning of different types of IFNs. To investigate the receptor system of IFN-υ, the change in the expression of ISGs induced by the overexpression of IFN-υ was detected in zebrafish when all class II cytokine receptors, except *il22bp* (CRFB9) and the

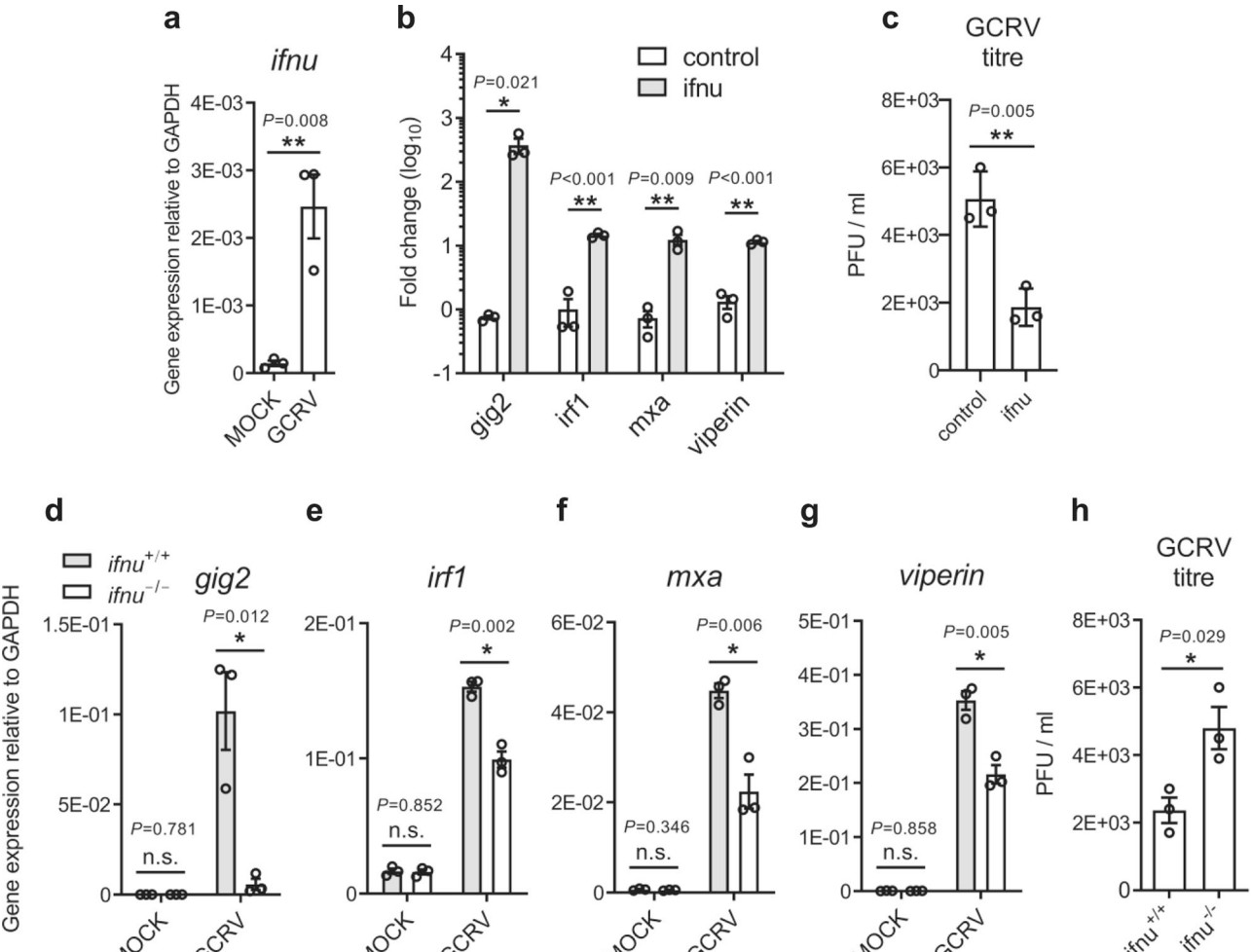

**Fig. 2 Antiviral function of zebrafish IFN-υ. a** Induction of IFN-υ gene by GCRV. Zebrafish larvae (5 dpf, $n = 33$) were infected with GCRV for 24 h and were collected to extract RNA to determine the expression of *ifnu*, which was normalized against *gapdh*, by using quantitative RT-PCR. **b** Induction of antiviral ISGs in IFN-υ-stimulated zebrafish. Embryos ($n = 150$) were injected at one-cell stage with IFN-υ or empty vector plasmid, and after 72 h, the mRNA level of ISGs was detected by quantitative RT-PCR. The expression of the selected genes was normalized against *gapdh* and fold changes were calculated relative to control group (empty vector). **c** Analyses of viral titers in IFN-υ-stimulated zebrafish. Embryos ($n = 150$) at one-cell stage were expressed with IFN-υ plasmid or empty vector for 72 h, and the hatched zebrafish larvae ($n = 33$) were infected with GCRV for 24 h and then collected to detect viral titers. Effects of IFN-υ deficiency on ISG expression, such as *gig2* (**d**), *irf1* (**e**), *mxa* (**f**) and *viperin* (**g**), and viral titers (**h**) in response to GCRV infection. 5 dpf zebrafish ($n = 33$) were infected with GCRV for 24 h and then were collected to determine ISG expression and viral titer. The expression of the selected genes was normalized against *gapdh*. Data are expressed as mean ± SEM from three independent experiments. The two-tailed Student's *t* test was used to determine the statistical significance, with * indicating $P < 0.05$, and ** indicating $P < 0.01$.

duplicated Tissue Factor genes (CRFB10 and 11), were separately knocked down using the specific morpholino oligonucleotides for receptor genes, which have been proved previously to possess knockdown activity[10,20,45]. The knockdown effect of these morpholino oligonucleotides used in this study was also examined (Supplementary Fig. 4a–i), and the expression of ISGs induced by type I or II IFNs was significantly inhibited by specific morpholinos, respectively (Supplementary Fig. 4a, b). The transmembrane receptor knockdown results showed that the existence of CRFB12 and CRFB4 (IL-10RB) morpholino, but not the morpholino oligonucleotides for other receptors (such as CRFB1, 2, 5, 6, 7, 8, 13, 14, 15, 16 and 17), in the IFN-υ-treated zebrafish embryos at 24 and 48 h post-fertilization (hpf) always blocked the expression of the inducible ISGs, including *gig2*, *irf1*, *mxa* and *viperin* (Fig. 3 and Supplementary Fig. 5). Moreover, CRFB12 gene (*ifnur1*) was found to have high expression level in zebrafish embryos (Supplementary Fig. 6a, b), suggesting that IFN-υ is involved in establishing antiviral immunity in zebrafish embryos.

Subsequently, to verify CRFB12 and CRFB4 are involved in IFN-υ functions, *crfb12* and *crfb4* knockout zebrafish were generated by Cas9/gRNA system. The 56 bp and 5 bp deletion in exon3 of CRFB12 and CRFB4 genes respectively, both led to premature translation termination (Supplementary Fig. 7a–d). It was observed that the induction of ISGs and anti-GCRV activity by IFN-υ were abolished in *crfb12* and *crfb4* deficiency zebrafish compared to the WT (Fig. 4a, b).

Marked developmental defects were not observed in *crfb4* knockout zebrafish (Supplementary Movie 1), and also not in *crfb12* and *ifnu* deficiency fish (Supplementary Fig. 8a–c, and Supplementary Fig. 9a–c). But, the knockdown of *crfb4* was lethal to zebrafish, as reported previously[45] (Supplementary Figs. 10 and 11). Although the knockdown and knockout of *crfb4* resulted in different phenotypes in relation with embryonic development in zebrafish, the removal of both CRFB12 and CRFB4 through knockdown or knockout all impaired the expression of antiviral ISGs as induced by IFN-υ. It is thus

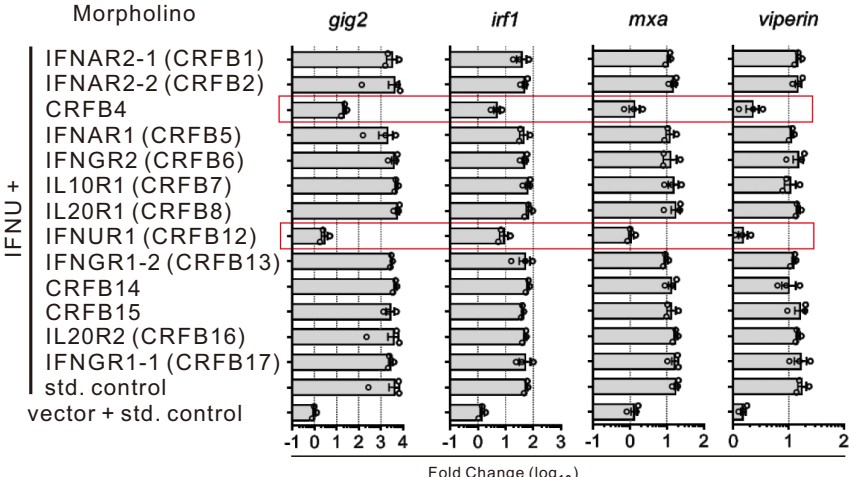

**Fig. 3 Effects on zebrafish IFN-υ-induced ISG expression by knockdown of CRFB morpholinos.** IFN-υ expressing plasmid and different morpholinos were co-injected into zebrafish embryos ($n = 500$) at one-cell stage and 48 h later the expression of ISGs was detected by quantitative RT-PCR. The expression of the selected genes was normalized against *gapdh* and fold changes were calculated relative to control group which was co-injected in embryos ($n = 150$) with empty plasmid (i.e., vector) and standard (std) control morpholinos.

suggested that CRFB12 and CRFB4 are involved in IFN-υ-mediated signaling. As the ligand of CRFB12 in fish has not been verified so far since its identification, CRFB12 should then be designated as IFN-υR1 (*ifnur1*).

**Evolutionary analysis of *IFNU* in lineages of vertebrates.** To determine whether *IFNU* and *IFNUR1* orthologues may exist in other vertebrates, we performed BLAST and synteny analyses based on the available vertebrate genome data from public database. It is discovered that *IFNU* gene with conserved five-coding exons exists widely in vertebrates, including cartilaginous fishes, teleosts, amphibians, reptiles, birds and mammals, with the intron phases of all *IFNU* genes being zero (Supplementary Table 2 and Supplementary Fig. 12). Multiple sequence alignments showed that vertebrate IFN-υ contains conserved sequence features, such as the cysteine and tryptophan/leucine pattern (CXXXXX[W/L]; Supplementary Fig. 12) at N-terminal region. Sequence alignments of IFN-υ with members in other three types of IFNs revealed that IFN-υ in zebrafish, clawed frog, green anole, chicken, and platypus (*Ornithorhynchus anatinus*) have low protein sequence identities, being 5.5–17.1%, 2.6–13.6%, 7.4–15.5%, 11.1–15.1% and 4.8–19.0%, with all other types of IFNs in these animals, respectively (Supplementary Figs. 13–17). Furthermore, protein structures of IFN-υ and other types of IFNs from zebrafish, clawed frog, green anole, chicken and platypus were predicted by using AlphaFold 2, except zebrafish IFNphi1 (PDB: 3piv) and IFNphi2 (PDB: 3piw) which have been characterized crystallographically[21]. Structural comparison showed that the integral structure of these IFN-υ proteins from zebrafish to platypus resembles more their own type I IFNs with root mean square deviation (RMSD) value being 4.167–7.633 Å (across all pruned atom pairs) than type II (RMSD, 19.830–29.279 Å) or III IFNs (RMSD, 8.701–11.533 Å), respectively (Supplementary Figs. 18 and 22). In addition to the similar structure at C-terminal region, the N-terminal region between type I IFN and IFN-υ could not be well overlaid, with the corresponding Helix A region representing the noticeable difference (Supplementary Figs. 18 and 22). It is thus indicated that IFN-υ gene is obviously distinct from type I, II and III IFNs in consideration of sequence and putative structure.

*IFNUR1* genes were also identified in vertebrates from fish to mammal, and the intron phases of most IFN-υR1 genes are

conserved, being 1-2-1-0-1-0, when compared with receptor genes of other IFNs[9] (Supplementary Fig. 23). IFN-υR1 consists of conserved structures of class II cytokine receptors, such as signal peptide, extracellular (with a D200 domain containing two FNIII-like domains), transmembrane and intracellular regions (Supplementary Fig. 23). Most sequences related to the potential beta-sheet structures in IFN-υR1 extracellular region are conserved when compared with other IFN receptors from fish to mammal (Supplementary Fig. 24). On the other hand, alignments showed that membrane-proximal region of IFN-υR1 intracellular sequence possesses a potential docking site for JAK family members, including a conserved 'box1' membrane-proximal receptor peptide motif with PXXL sequence and a hydrophobic residues-rich 'box2' receptor motif (Supplementary Fig. 25). In fact, the 'box1' and 'box2' sequences of class II cytokine receptors have been proved as the critical motifs for binding JAK family members[46]. Moreover, two highly conserved tyrosine sites were found in IFN-υR1 intracellular regions from fish to mammal, which may be associated with STAT activation (Supplementary Fig. 25).

Syntenic analysis revealed that IFNU loci are highly conserved from elephant shark (*Callorhinchus milii*) to platypus, with the presence of a single copy of IFNU gene located between adenosine deaminase, RNA specific B2 (inactive) (*ADARB2*) and phosphofructokinase, platelet (*PFKP*), except in elephant shark which has two IFNU genes (Fig. 5a, and Supplementary Fig. 26a). The IFNU locus is different from all known IFN loci, i.e., type I, type II and type III IFN loci, which are linked with 3-hydroxyacyl-CoA dehydratase 4 (*HACD4*)/rho GTPase activating protein 27 (*ARHGAP27*), dual specificity tyrosine phosphorylation regulated kinase 2 (*DYRK2*) and syncollin (*SYCN*)/SPT5 homolog (*SUPT5H*), respectively (Fig. 5b, c, d).

IFNUR1 genes are also identified in a unique and conserved locus from different species of vertebrates (Supplementary Fig. 26b, and Supplementary Fig. 27). However, we could not identify IFN-υ or IFN-υR1 orthologue(s) in human genome, even on conserved *IFNU/IFNUR1* locus in vertebrates (Fig. 5a, and Supplementary Fig. 27). In fact, *IFNU* and *IFNUR1* genes are only found in species in the Monotremata, such as platypus and Australian echidna (*Tachyglossus aculeatus*) (Supplementary Fig. 26, and Supplementary Table 2), but not in Metatheria and Eutheria species, such as common brushtail (*Trichosurus vulpecula*), Tasmanian devil (*Sarcophilus harrisii*), pig (*Sus scrofa*)

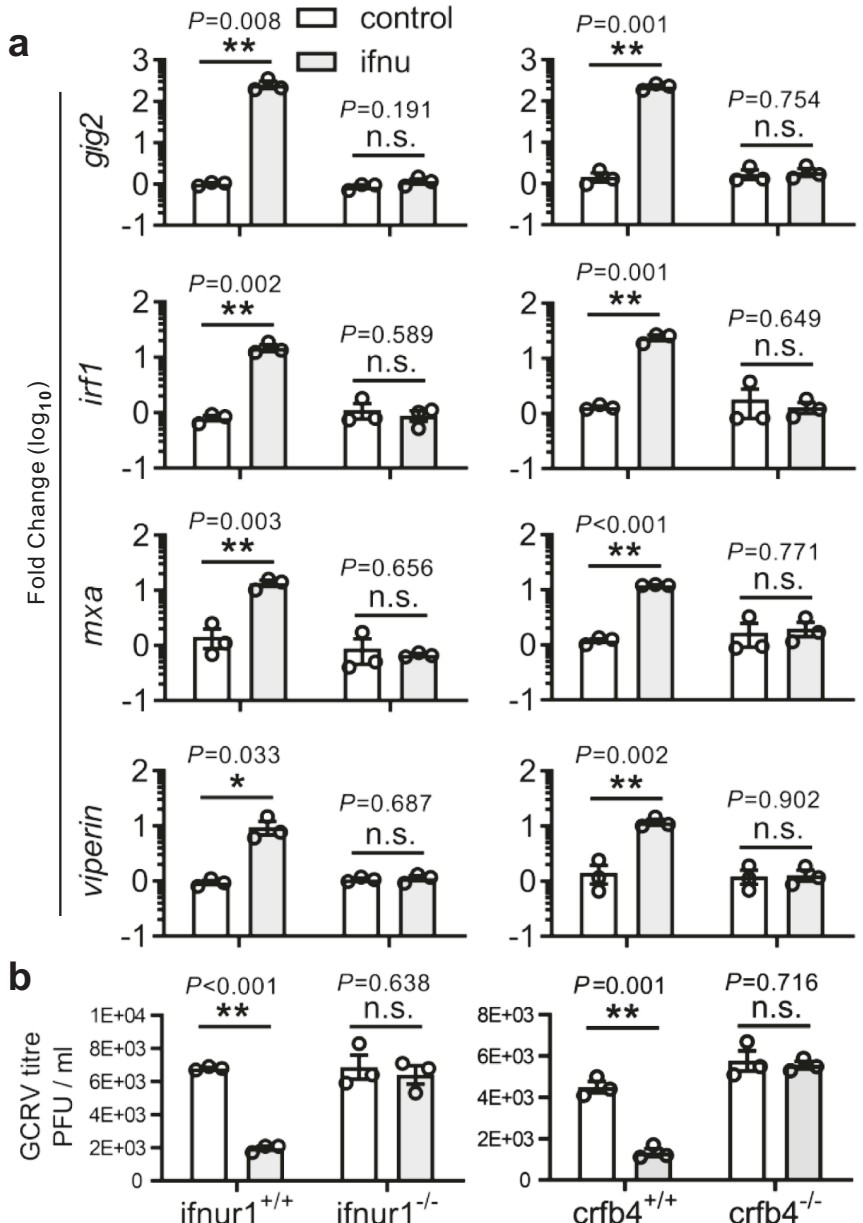

**Fig. 4 Knockout effect of *ifnur1* and *crfb4* on zebrafish IFN-υ signaling.** Analyses of ISG expression (**a**) and viral titer (**b**) in *ifnur1* and *crfb4* deficiency and WT zebrafish stimulated by IFN-υ. Embryos ($n = 150$) at one-cell stage were micro-injected with IFN-υ or control plasmids for 72 h, and hatched zebrafish larvae ($n = 33$) were collected for quantitative RT-PCR detection or infected with GCRV for 24 h to determine viral titer. The expression of ISGs was normalized against *gapdh*. Data are expressed as mean ± SEM from three independent experiments. The two-tailed Student's *t* test was used to determine the statistical significance, with * indicating $P < 0.05$, and ** indicating $P < 0.01$, n.s. non-significancy.

and Arabian camel (*Camelus dromedarius*) (Supplementary Fig. 26).

To illustrate the evolutionary relationship of *IFNU* and *IFNUR1* genes in vertebrates, protein sequences of class II cytokines were used to construct phylogenetic trees. In general, vertebrate IFN-υ, IL-10-related cytokines, type I, II and type III IFNs are grouped into separate clades (Fig. 6), and vertebrate IFN-υR1 genes are also clustered together to form a unique clade (Supplementary Fig. 28). It is suggested that IFN-υ/IFN-υR1 and other IFN/IFN receptor genes might have diverged before the appearance of fish.

Furthermore, *ifnu* (GenBank accession no.: MW924834), IFN-υR1 and IL-10R2 (homologous to CRFB4) were identified from clawed frog (*Xenopus laevis*) A6 cell line to verify the function of IFN-υ in amphibian. IFN-υ also shares low identity with other class II cytokines including type I IFNs (11.2–13.7%), type II IFN (8.3%), type III IFNs (14.3–14.8%) and IL-10 (8.3%) in clawed frog. Recombinant clawed frog IFN-υ protein induced significantly the expression of ISGs (*mx1* and *viperin*) and reduced the titer of frog virus 3 (FV3) in A6 cells (Supplementary Fig. 29a−c). However, antiviral activity of IFN-υ was impaired by knockdown of IFN-υR1 and IL10R2 (Supplementary Fig. 29d−h).

## Discussion

IFNs are a subset of class II cytokines, which share sequence and probably functional properties in having multi-alpha helices, and in inducing antiviral genes to interfere with viral replication in hosts[12,47]. In the present study, a formerly uncharacterized gene of class II cytokines, designated as IFN-υ (*ifnu*), was identified

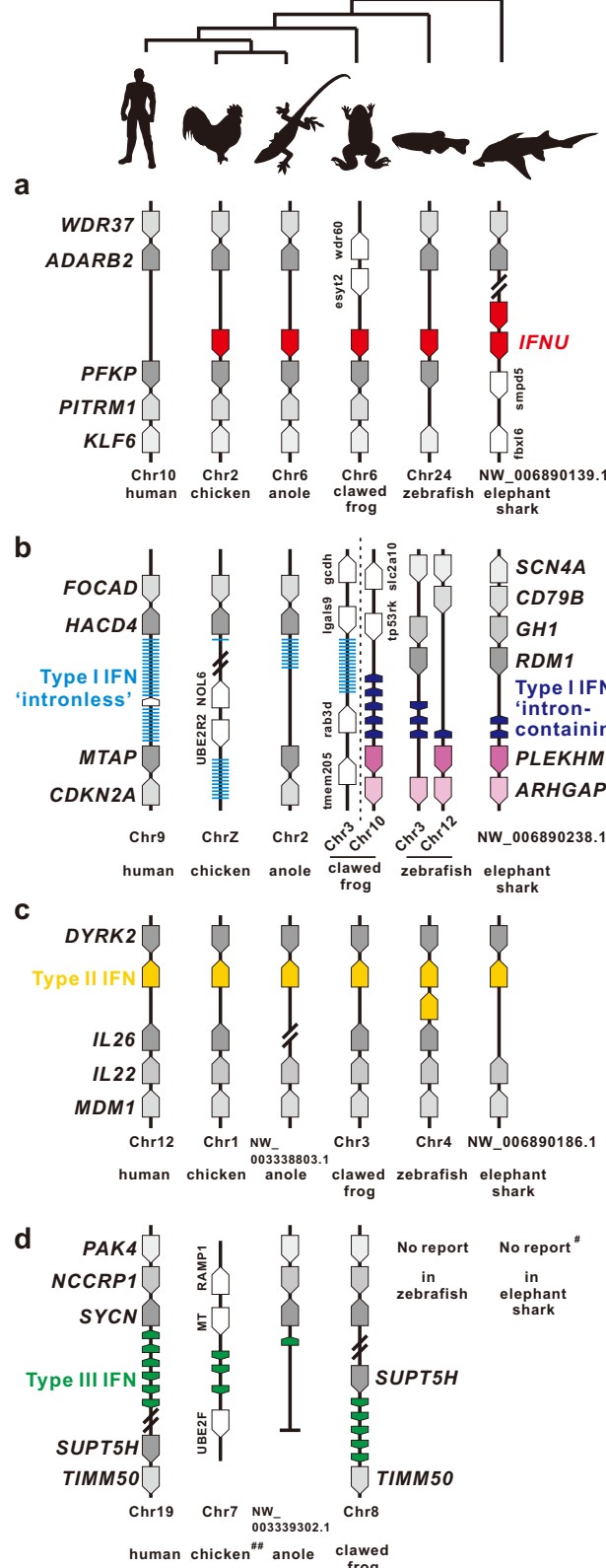

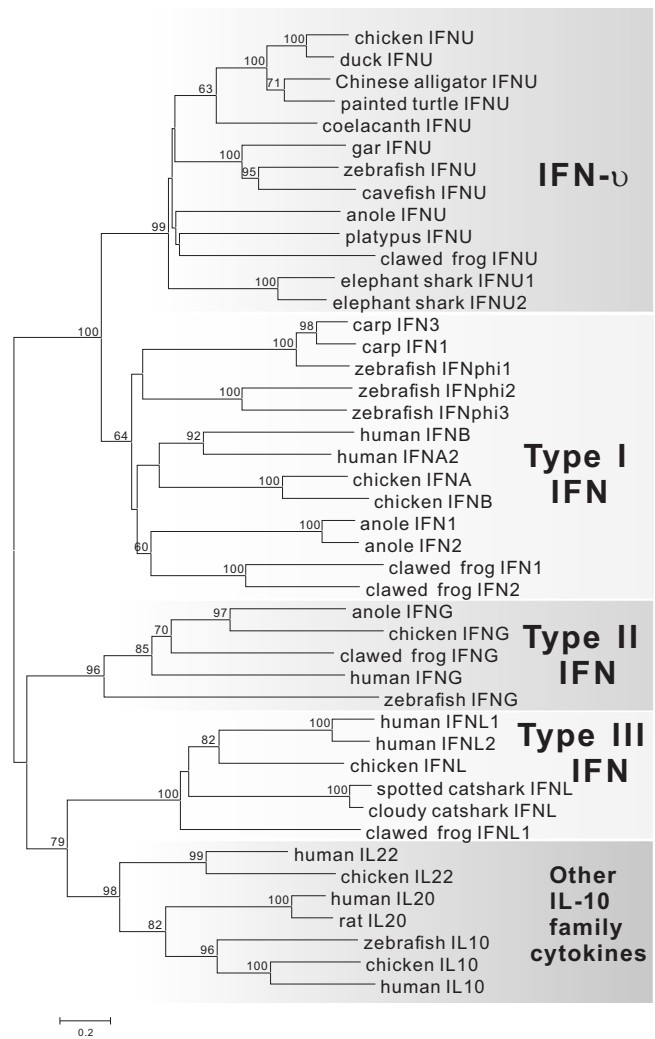

**Fig. 5 Gene synteny of type I to IV IFNs.** Gene synteny of *IFNU* (**a**), type I (**b**), II (**c**) and III IFN (**d**) loci in vertebrates. All genes are indicated with arrow symbols which point to the transcription direction. *IFNU*, type I, II and III IFNs are dyed using red, blue, yellow and green, respectively. The conserved and non-conserved neighbor genes are illustrated in gray and white, respectively. Evolutionary topology is supported by a recent study[35]. ♯ indicating no reports on type III IFN in elephant shark, but in catshark[8], and ♯♯ chicken type III IFN locus which belongs to the conserved "IFNLB loci" in reptile and bird[34].

**Fig. 6 Phylogenetic analysis of *IFNU* in vertebrates.** Protein sequences from class II cytokines in vertebrates were used to construct the neighbor-joining (NJ) tree. Sequence accession numbers are listed in Supplementary Table 5.

and functionally characterized in zebrafish and clawed frog. IFN-υ shares several typical features with known IFNs. Firstly, this IFN-υ protein in zebrafish as well as in other vertebrates has a similar number of amino acids being about 200 with other known IFN proteins (about 200 aa), and has a signal peptide and six putative alpha helices, as type I, type II and type III IFNs with five to six helices, respectively. Secondly, IFN-υ genes share same

genomic organization with intron-containing type I IFN genes in fish and frog and intron-containing type III IFNs in tetrapod, which are mainly encoded by five-coding exons[15,18,20,34]. Thirdly, IFN-υ needs class II cytokine receptors, IFN-υR1 and CRFB4 (IL-10R2 in clawed frog), as other three types of IFNs all use this category of receptors for their signaling[3,7,15]. Fourthly, IFN-υ is able to regulate the expression of ISGs, such as *gig2*, *mxa/mx1*, *irf1* and *viperin*, which are important antiviral effectors as induced by other IFNs[7,31,48,49]. In particular, virus replication can be markedly inhibited by IFNs, especially by type I and type III IFNs[31], and as observed in the present study GCRV and FV3

replications were inhibited by IFN-υ, and such effect should be the well-known functional feature of IFNs[12,47,50].

IFN-υ orthologues were found in different lineages of vertebrates, with conserved sequence feature, such as the type I IFN-like C-terminal sequences, which is distinguished obviously from type II and III IFNs. However, genomic organization of multi-exons in vertebrate (including amniotes) of IFN-υ is similar to the intron-containing type I IFN genes in fish and frog and intron-containing type III IFNs in cartilaginous fish and tetrapod, but not the intronless type I IFNs in amniotes. In fact, IFN-υ is located at a unique and highly conserved locus from fish to primitive mammals, being distinct from all the three-type IFNs[15–18,24,27,34]. Thus, the co-existence of IFN-υ and all three-type IFNs in cartilaginous fish, amphibian, reptile, avian and primitive mammals may also indicate the uniqueness of this currently identified IFN in vertebrates.

It has been well documented that the reported three types of IFNs, i.e., type I, type II and type III IFNs are interacted separately with different receptors, and are distributed on different loci, and these IFNs are phylogenetically clustered into separate clades[3,15,24,31]. In vertebrates, all type I IFNs, including those representing the evolutionary continuum of type I IFNs, or those diversified independently in amphibians, use the receptor subunits, IFNAR1 and IFNAR2[12,15,17,51], although in teleost fish, IFNAR2 genes are replicated to IFNAR2-1 (CRFB1) and IFNAR2-2 (CRFB2), which are associated with the two groups of type I IFNs, respectively[7,15,20].

It is unique that type II IFN in tetrapods has only a single member, IFN-γ, and is evolutionarily distributed in a conserved locus, which all uses the receptor complex, IFNGR1 and IFNGR2[24,27]. But in teleost fish, two members, IFN-γ and IFN-γrel, have been reported as type II IFNs[28]. Phylogenetic analysis supported that the replicated receptor subunits, IFNGR1-1 (CRFB17) and IFNGR1-2 (CRFB13) in fish are homologous to tetrapod IFNGR1, and it is verified that IFNGR1-1, IFNGR1-2 and IFNGR2 (CRFB6) serve as the receptors for the type II IFNs in teleost fish[7,10,15].

Type III IFNs have been reported in cartilaginous fish, amphibian, reptile, avian and mammals[8,19,33,34,52–54], with the conservation in gene locus and ligand-receptor system (IFNLR1 and IL-10R2), except that the location of IFNL gene on chromosome is still unclear in cartilaginous fish[8].

In mammal, the critical helix A of all three-type IFN mature peptides is involved in the interface formation of IFNs and the specific ligand-binding chain[55–58]. It is noteworthy that the N-terminal region of IFN-υ shows low identity with the known class II cytokines in zebrafish, implying that IFN-υ does not share the ligand-binding chain with the known IFNs. Indeed, the functional receptors of IFN-υ identified in the present study are IFN-υR1 and IL-10R2/CRFB4, which are different from the receptors of type I, type II, and type III IFNs, although IL-10R2/CRFB4 is likely shared by IFN-υ and type III IFNs as a chain of their receptor complex.

Furthermore, some ligands and receptors of type I or II IFNs were duplicated in teleost due to the teleost-specific whole-genome duplication (TGD)[15]. Nevertheless, in phylogenic trees, the fish duplicated IFNs and IFN receptor genes were always grouped firstly with the corresponding homologous genes in other vertebrates[8]. In this study, fish IFN-υ and IFN-υR1 genes were clustered firstly with their own homologous genes in other vertebrates. Therefore, the independent phylogenic relationship of IFN-υ as well as IFN-υR1 in vertebrates also suggests this unique IFN, and the uncovered role of this receptor, in vertebrates.

It is certainly interesting to discuss the evolutionary presence of IFN-υ gene in the Theria, including marsupials and placentals. Despite the presence of IFN-υ in highly conserved ADARB2-PFKP locus in vertebrates as discussed above, IFN-υ is missing in human genome, and cannot be found on the collinear locus of other mammals, even with relatively good quality assembly of their genomes, in the Metatheria and Eutheria, such as Tasmanian devil (Sarcophilus harrisii), yellow-footed antechinus (Antechinus flavipes), gray short-tailed opossum (Monodelphis domestica), common brushtail (Trichosurus vulpecula), monito del monte (Dromiciops gliroides), Cape golden mole (Chrysochloris asiatica), pig (Sus scrofa), Arabian camel (Camelus dromedarius), African savanna elephant (Loxodonta africana) and mouse (Mus musculus). Interestingly, IFN-υR1 gene, as IFN-υ, is also found only in species from the Monotremata (egg-laying mammals), such as platypus, but not in the Theria. In fact, the gene arrangement of CLIC2-FAAH2 locus, where IFN-υR1 is located in platypus, is inverted in Theria species when compared with platypus. It is thus considered that chromosome rearrangement might have resulted in the gene loss of IFN-υR1 in Theria species under selective pressure, as the gene loss of a receptor or a ligand may cause specifically the loss of the ligand-receptor system in evolution[59]. However, this may provide an indirect evidence to support the ligand-receptor system of IFN-υ and IFN-υR1.

It has been reported that class II cytokines and class II cytokine receptors might have undergone duplication to form specialized distinct ligand-receptor systems[60], and this may represent an evolutionary foundation for functional diversification of genes[61]. In this study, IFN-υ gene shows the similar C-terminal sequence as type I IFNs, and shares the signal-transduction receptor, IL-10R2 as type III IFNs, which is also shared among some IL-10 family cytokines[2]. Thus, it may be implied that IFN-υ and type I/III IFNs share a common ancestor. In cartilaginous fish, the coexistence of type I, III IFN and IFN-υ genes is observed, indicating that these three types of IFNs might have diverged before the appearance of cartilaginous fish, and IFN-υ gene and type I/III IFNs may have undergone independent evolution to form specific ligand-receptor pairs, including IFN-υ/IFN-υR1, type I IFN/IFNAR2, and type III IFN/IFNLR1. Therefore, it can be hypothesized that the loss of IFN-υ gene in the Theria might be resulted from the lack of IFNUR1 due to the chromosome rearrangement, which may, on the other hand, support the ligand-receptor system of IFN-υ and IFN-υR1. In addition, intronless type I IFNs in amniotes might have originated from retroposition of intron-containing type I IFN transcripts in lower vertebrates[17,19,62]. Mammalian intronless type I IFNs may have undergone further expansion to form a few more subtypes, but some of them do not exist in all mammal lineages, such as IFN-ε and IFN-τ, which are found in placentals but not in marsupials[63]. IFN-υ gene might have been lost possibly before the divergence of placentals, and it seems likely that IFN-υ gene and type I IFN subtypes may have undergone independent evolution in mammals.

In consideration of the sequence feature, the gene locus, the phylogeny, the receptor composition, it is reasonably concluded that IFN-υ is a unique member of class II cytokine family, in addition to the presence of type I, type II or type III IFNs in vertebrates.

## Methods

**Ethics declaration**. All animal experiments were conducted in accordance with the Guiding Principles for the Care and Use of Laboratory Animals in the Institute of Hydrobiology, Chinese Academy of Sciences, and were approved by the Animal Ethic and Welfare Committee in the Institute of Hydrobiology.

**Data mining and sequence analysis**. Due to the low expression level and relative short coding exons, the following strategies were used to scan the available genomic sequences of zebrafish. First, unannotated sequences (or the so-called intergenic region in the annotation version of Zv9) between annotated genes in zebrafish

genome (assembly version: Zv9) were collected to carry out gene prediction or annotation using GENSCAN and FGENESH programs with standard parameters[64,65], as in previous studies on the discovery of type III IFNs from human genomic sequence data[52,53]. Second, the predicted genes were screened based on class II cytokine gene features, including 1) multi-exon organization, usually with five-coding exons, 2) intron phase being zero, and 3) putative protein sequence containing signal peptide at N-terminal region. Third, multiple sequence alignments and PSIPRED program (http://bioinf.cs.ucl.ac.uk/psipred/) were used to predict alpha-helix regions. For the identification of IFNU and IFNUR1 genes in other vertebrates, BLAST and FGENESH programs were used to analyze available genome sequences from different species which were shown in Supplementary Table 2. For sequence analysis, Signal peptides and transmembrane regions were predicted using the program SignalP 3.0 Server (http://www.cbs.dtu.dk/services/SignalP-3.0/) and TMHMM Server v. 2.0 program (http://www.cbs.dtu.dk/services/TMHMM-2.0/), respectively. Protein structures were predicted by AlphaFold 2 and visualized by using Chimera (Version 1.15).

**Phylogenetic analysis**. Multiple sequence alignments were performed using the Clustal X program. The predicted intracellular region sequences of class II receptor protein at C-terminus were removed manually since the extracellular region of receptor is mainly involved functionally in ligand interaction[1,52]. Neighbor-joining (NJ) phylogenetic trees were constructed using MEGA4 package with 1000 time repeat of bootstrap analysis.

**Experimental animal, virus and cell**. The zebrafish (AB strain) embryos and adults were purchased from China Zebrafish Resource Center (CZRC) and maintained at 28.5 °C with 10 h dark/14 h light cycle according to a previously study[66].

Grass carp reovirus (GCRV) is a dsRNA virus, belonging to the genus of *Aquareovirus* in the Reoviridae, and causes severe economic loss in aquaculture of China[67,68]. GCRV can cause the infection and immune response in some other cyprinid fishes, including zebrafish and another model fish used for toxicity research, rare minnow (*Gobiocypris rarus*) etc.[69,70]. In fact, GCRV has been a model for studying reovirus 3 D structure, cell-entry mechanism and antiviral immunity[71]. Thus, GCRV is used to investigate the antiviral function of zebrafish IFN-υ.

For cell culture, *Ctenopharyngodon idella* kidney (CIK) cells were purchased from the China Center for Type Culture Collection (CCTCC) and were cultured in minimum essential media (MEM, gibco®, Thermo Fisher Scientific) supplemented with 10% fetal bovine serum (FBS, gibco®, Thermo Fisher Scientific). Cells were maintained at 28 °C in an incubator (HERACELL 150i, Thermo Fisher Scientific) with 5% CO2. GCRV was grown in CIK cells maintained in MEM with 2% FBS as described previously, and supernatants from the virus infected cells were separated by centrifugation[68,72]. Viral titer was measured by plaque assay[68,72]. CIK cells (about $1.0 \times 10^6$ cells per well) were seeded in sterile 24-well plate, before being infected with a series of tenfold dilutions of GCRV in FBS-free MEM. Two hours post-infection (hpi), the FBS-free MEM with GCRV was replaced with fresh MEM containing 2% FBS and 1% methylcellulose. At 72 hpi, the medium was removed, and cells were fixed with 10 formaldehyde solution for staining using 5% crystal violet solution (Sigma-Aldrich) to count plaque-forming units (PFUs). Experiments were repeated two times.

The *Xenopus laevis* A6 cells (CCL-102) were obtained from American Type Culture Collection (ATCC) and cultured in 75% NCTC-109 medium (gibco®, Thermo Fisher Scientific) with 15% distilled water and 10% FBS in a 5% CO2 incubator. Human embryonic kidney 293T (HEK293T) and *Epithelioma papulosum cyprini* (EPC) cells were cultured in DMEM (gibco®, Thermo Fisher Scientific) with 10% FBS at 37 °C with 5% CO2 and medium 199 (gibco®, Thermo Fisher Scientific) with 10% FBS at 28 °C with 5% CO2, respectively. Frog virus 3 (FV3, DNA virus) from ATCC was propagated in EPC cells and the virus titer was determined by standard plaque assays as previously reported[17]. Briefly, EPC cells were seeded in sterile 24-well plate at 25 °C with 5% CO2 overnight. After washing by FBS-free MEM, EPC cells were incubated with FV3 in FBS-free MEM for two hours. Then the supernatants were removed and replaced with fresh MEM containing 2% FBS and 1% methylcellulose. EPC cells were fixed and stained using formaldehyde and crystal violet solution, respectively, to count PFUs when the plaques were exposed.

**Total RNA preparation, cDNA synthesis, gene cloning and plasmid construction**. Total RNA was extracted from zebrafish tissues, larvae or A6 cells using TRIzol® Reagent (Ambion) and was purified with UNIQ-10 Column Trizol Total RNA Isolation Kit (Sangon Biotech) following manufacturers' protocols. DNase I (RNase-free, Thermo Fisher Scientific) was purchased to digest the DNA in isolated total RNA before the synthesis of first-strand cDNA using RevertAid™ First Strand cDNA Synthesis Kit (Thermo Fisher Scientific) with the mixture of oligo(dT)18 and random hexamer nucleotides according to the manufacturer's instruction.

Primers for RACE-PCRs (Supplementary Table 3) were designed on the basis of predicted transcripts of zebrafish IFN-υ gene using Primer 5.0 software. 5-end and 3-end RACE PCRs were performed to generate IFN-υ fragments using Ex-taq PCR

amplification system (Takara) and RACE cDNA template synthesized by GeneRacer™ Kit (Invitrogen) following the manufacturers' instructions. The RACE PCR program was: one cycle of 94 °C for 3 min; 8 cycles of 94 °C for 30 s, 65 °C for 30 s and 72 °C for 2 min; 28 cycles of 94 °C for 30 s, 62 °C for 30 s and 72 °C for 2 min. The PCR products were then subcloned into pMD18T vector (Takara) for sequencing.

The full-length coding sequences of *ifnu*, *ifnur1*, and *crfb4* in zebrafish and A6 cells were amplified from cDNA templates using specific primers (Supplementary Table 3) and were subcloned separately into p3XFLAG-CMV-14 expression vector (Sigma-Aldrich) or pMD18T vector. The constructed plasmids were extracted and purified by using E.Z.N.A.® Endo Free Plasmid Mini Kit II (Omega Bio-tek) according to the manufacturer's instruction, and the plasmids were also sequenced for verification.

**Morpholino and microinjection**. The gene-specific morpholinos (MOs) against CRFBs were reported previously[10,20,45] (Supplementary Table 4), and all MOs were obtained from Gene Tools and were dissolved in sterile nuclease-free water. Microinjection was performed as reported in other studies concerning IFN research in zebrafish[10,29,45], with some modification. Briefly, expression plasmids at a concentration of 100 ng/μL with 0.1% phenol red as an indicator were injected alone or co-injected with MOs at a concentration of 1 mM into zebrafish embryo at one-cell stage using PLI-100A Pico-Injector (Harvard Apparatus). The control groups were injected with empty vectors alone or co-injected with standard control MO (1 mM). Each embryo was injected with 2 nL solution.

**Gene knockout**. The knockout line of zebrafish *ifnu*, *ifnur1* and *crfb4* was generated using Cas9/gRNA system which was described previously in zebrafish[29]. The single-guide RNAs (sgRNAs) for *ifnu*, *ifnur1* and *crfb4* were designed using the software online (http://crispr.mit.edu). The zebrafish codon optimized Cas9 plasmid (pGH-T7-zCas9) was linearized by *Xba* I restriction enzyme, and Cas9 RNA was synthesized and purified using mMESSAGE mMACHINE™ T7 ULTRA Transcription Kit (Ambion). GRNA-PMD19-T plasmids were used as the template to amplify *ifnu*, *ifnur1* and *il10r2* sgRNA templates by using primer pairs, including IFNUsgRNA-F + sgRNA-R, crfb12-E3-sgRNA1-F + sgRNA-R and Crfb4-B3 + sgRNA-R, respectively (Supplementary Table 3). The sgRNA was synthesized using MEGAshortscript™ T7 Transcription Kit (Ambion). The genomic DNA was extracted from 30 embryos for mutant detection by sequencing of PCR products generated by primers, IFNUmut-F/IFNUmut-R, crfb12-E3-F1/crfb12-E3-R1 and Crfb4-F2/Crfb4-R2 (Supplementary Table 3) respectively, at 48 h post-injection. The remainders of positive embryos were raised to adulthood as the F0, which were backcrossed with wildtype (AB line) zebrafish to generate F1. The target sites of F1 were detected by PCR and DNA sequencing. The heterozygous F1 was backcrossed with wildtype (AB line) to generate F2, which with the same mutant was intercrossed to generate F3, including WT (*ifnu*[+/+], *ifnur1*[+/+], and *crfb4*[+/+]) and homozygous mutants (*ifnu*[−/−], *ifnur1*[−/−] and *crfb4*[−/−]).

**Fish infection, receptor knockdown and antiviral activity of zebrafish IFN-υ**. The knockdown effects of the CRFB MOs were evaluated by semi-quantitative PCRs or expression changes of ISGs induced by type I and II IFNs using qRT-PCR with the specific primers (Supplementary Table 3) as previously described[10,20]. In brief, CRFB/control MOs were injected alone or co-injected with IFN expressing/empty plasmids into zebrafish embryos at one-cell stage. 48 h later, the embryos were collected to extract total RNA for further analyses.

For zebrafish larvae infection, 11 individuals as one group were maintained in egg water by filter sterilization using 0.22 μm Syringe-driven Filter Unit (Merck Millipore) and were infected with GCRV at a final concentration of about $2.0 \times 10^7$ PFU/mL by immersion in sterile plates (Thermo Fisher Scientific) at 25 °C. All experimental animals were sacrificed by ice-bath anaesthetizing and then were collected for further analyses at 24 hpi.

To detect viral titer, 33 GCRV-infected individuals from the same treated zebrafish larvae were collected into one sterile tube and washed with filtered sterile water six times. Subsequently, the sacrificed animals were homogenized in 1 mL sterile MEM without FBS, and then centrifuged for the collection of supernatant, which was then filtered through 0.22 μm Syringe-driven Filter Unit (Merck Millipore) and used to determine the titer according to the protocol described above. Experiments were repeated three times.

**Recombinant protein, receptor knockdown and antiviral activity of clawed frog IFN-υ**. *X. laevis* IFN-υ expression plasmids and empty vectors (as control) were transfected into HEK293T cells by Lipofectamine 2000 (Invitrogen) following the manufacturer's protocols to generate recombinant flag-tag-IFN-υ protein, which was detected by Western blotting. Expression knockdown of IFN-υR1 and IL10R2 (CRFB4) in A6 cells were achieved by RNA interference, and cDNA sequences of siRNA targeting IFN-υR1 and IL10R2.L/S mRNA sites are 5-CCTCATTCTTACTGTTACT-3 and 5-CATCTGAAAGAGTACCTAA-3, respectively. All siRNA, including a negative control siRNA, were designed and synthesized by RiboBio Co., Ltd and were transfected into A6 cells using X-tremeGENE siRNA Transfection Reagent (Roche) according to the manufacturer's instruction. The silencing effects of the IFN-υR1 and IL10R2 siRNAs

were detected by qRT-PCR using the specific primers (Supplementary Table 3). Untreated or siRNA-transfected A6 cells (at 72 h post-transfection, hpt) were incubated with recombinant IFN-υ containing supernatant produced in HEK293T cells and control medium for 10 h in 48-well plate. Then, the cells were collected to analyze the expression of ISGs by qRT-PCR or were infected with FV3 at the multiplicity of infection (MOI) of 5 for antiviral assay.

**Quantitative real-time PCR.** The prepared cDNA from different samples were analyzed to examine the expression profile of genes using CFX96 Real-Time PCR Detection System (Bio-Rad). A final volume of 20 μL PCR reaction system was composed of 10 μL iQ$^{TM}$ SYBR$^®$ Green Supermix (Bio-Rad), 1 μL of each primers, 7 μL sterile water and 1 μL cDNA template, and the protocol was as the followings: one cycle of 95 °C for 3 min, followed by 45 cycles of 95 °C for 12 s, 60 °C for 20 s and 72 °C for 15 s. The gene expression for each sample was normalized against *gapdh* or *actb* (as an internal control). Data analysis was performed using the comparative Ct method ($2^{-\Delta\Delta ct}$ method). Specific primers for qRT-PCR were listed in Supplementary Table 3, including the previously reported primers for zebrafish *gapdh*, *irf1*, *mxa*, *gig2* and *viperin*, and A6 cell *actb*, *mx1* and *viperin*[17,20,73]. PCR products were sequenced to verify the specific amplification and were used as templates to determine the primer efficiency through standard curve amplification, which has been described previously[34].

**Antibodies and Western blotting.** Peptides (QASKNNNFNTTKC and CLKNSPSNETEPW) were synthesized and cross-linked with m-maleimidobenzoyl-N-hydroxysulfosuccinimide ester (MBS) and keyhole limpet hemocyanin (KLH) to immunize Wistar rats to raise polyclonal antiserum (anti-IFNUR1 and anti-CRFB4, respectively). Western blotting was carried out with reference to the previous report[34]. The total protein obtained from grinding and cell lysis buffer digestion of ice-anaesthetized zebrafish larvae, or supernatant medium from the IFN-υ expression plasmids or empty vectors transfected and untreated HEK293T cells, was separated by 10%/12% SDS-PAGE and then transferred to a polyvinylidene difluoride (PVDF) membrane (0.45 mm, Merck Millipore), which was incubated with antibodies (Dia-An Biotech, Inc), including anti-GAPDH (1:1000, Cat#: 2058, mouse monoclonal antibody, 2D7), anti-IFNUR1 (1:200, Cat#: C0814), anti-CRFB4 (1:200, Cat#: C0813) and anti-FLAG-tag (1:2000, Cat#: 2064, mouse monoclonal antibody, 2E5), in Tris-Buffered Saline with 0.05% (*v/v*) Tween 20 (TBST, pH 8.0) with 1% nonfat dry milk (*w/v*) at 4 °C overnight after blocking by 5% nonfat dry milk TBST for 3 h. After washing, the blot was incubated with secondary antibody (Merck Millipore). Chromogenic reaction was performed using Immobilon$^{TM}$ Western Chemiluminescent HRP Substrate (Merck Millipore) for staining the membrane and ChemiDoc$^{TM}$ MP imaging system (Bio-Rad) for examination.

**Whole-mount in situ hybridization.** Whole-mount fluorescence in situ hybridization was performed according to previous reports[74,75], with modification. Briefly, zebrafish embryos (24 hpf) were fixed in 4% of paraformaldehyde and were digested for 15 min using proteinase K (20 μg/ml) at 37 °C. After wash by PBS buffer, blocking of endogenous peroxidase by using 3% methanol-$H_2O_2$ and pre-hybridization were performed, and then the embryos were hybridized with the first probe (ifnur1 gene, Supplementary Table 6) or without probe (control) overnight at 42 °C. After washing with saline-sodium citrate buffer (SSC), the second DIG-labeled probe (Supplementary Table 6) was used to incubate the embryos. After wash in SSC, the sample was blocked by using rabbit serum for 30 min. Next, the embryos were incubated with anti-digoxigenin-labeled peroxidase (anti-DIG-HRP) for 40 min at 37 °C. After wash in PBS buffer, the embryos were incubated with Cy3-Tyramide signal amplification (TSA) and 4',6-diamidino-2-phenylindole (DAPI). The images were captured by using a fluorescence microscope (Leica M205FA).

**Statistical analyses.** Data were analyzed statistically with the two-tailed Student's *t* test in SPSS 16.0 software. Data are expressed as mean ± SEM, and significant difference is indicated with * $P < 0.05$, and ** $P < 0.01$.

**Reporting summary.** Further information on research design is available in the Nature Research Reporting Summary linked to this article.

## Data availability
The raw data for figures are provided in the Source data file. The complete CDS and amino acid sequence of ifnu in zebrafish, and also in African clawed frog have the GenBank accession numbers: MW547062.1, UBY00466.1, and MW924834.1, UBY00467.1. Source data are provided with the paper. Source data are provided with this paper.

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

## Acknowledgements

This work was supported by grants (Nos. 31320103913 (PN) and 31702369 (SNC)) from the National Natural Science Foundation of China (NFSC), by the China Agriculture Research System of MOF and MARA (CARS-46) (PN), and by special top talent plan "One Thing One Decision (Yishi Yiyi)" ([2018]27) (PN) and "First Class Fishery Discipline" program [(2018)8] (PN) in Shandong Province, China. We thank Dr. Kuoyu Li, Dr. Xunwei Xie, Dr. Luyuan Pan and Prof. Yonghua Sun in China Zebrafish Resource Center (CZRC) for assistance in zebrafish knockout experiments.

## Author contributions

P.N. led and supervised the studies. S.N.C. and P.N. designed experiments. S.N.C., J.H., Y.C.Y. and L.H. performed experiments and maintained zebrafish culture. S.N.C., B.H., S.W. and P.N. analyzed data. S.N.C. and P.N. performed bioinformatic analysis. S.N.C., Z.G. and P.N. wrote the manuscript.

## Competing interests

The authors declare no competing interests.

**Additional information**

