## [Peer Review File · Nature Communications]

Identification and establishment of type IV interferons and the characterization of interferon- γ including its class II cytokine receptors IFN- γ R1 and IL-10R2REVIEWER COMMENTS

Reviewer #1 (Remarks to the Author):

1. On page 5 under “Sequence characteristics of IFN-u”, the authors state that they deposited the full-length cDNA sequence for the zebrafish Ifnu gene in GenBank, but I cannot find this sequence in GenBank. Please provide the GenBank accession number for the sequence that you deposited.
2. The authors should also deposit and provide the amino acid sequence for their zebrafish IFN-u protein in GenBank and provide the corresponding accession number.
3. The Discussion section should include some text to explain why there is no IFN- ϵ gene in placental mammals.
4. The implication of transition from unique receptor chains for type I and type II IFNs to shared receptors for type III IFNs and other class-2 cytokines is interesting, especially because IFN-u retains the “type I IFN” C-terminus. The authors should discuss why IFN-u might represent a missing link between the type-I and type-III IFN gene families.
5. The sequence alignments shown in Fig. S4 did not include a broad range of species, and for the ligands, were not structurally based. A structure-based alignment of several type I IFNs, type II IFNs, type III IFNs and IFN- ϵ should be performed at the same time and shown in the same table.
6. Similar to my comment regarding Fig. S4, the sequence alignment of IFN- ϵ -R1 proteins in Fig. S5 should include an alignment of other class-2 receptor proteins, including IFN- α R1, IFN- α R2, IFN- λ R1 and IL-10R2.
7. There is no discussion as to why IFN- ϵ s are lost in placental mammalian genomes? The correlation between placental structure (and IFN- ϵ and IFN- τ being placental IFNs) and the loss of IFN-u needs some discussion here. Are they involved in establishing antiviral immunity in the laid egg? Are they expressed most strongly in embryos? An in situ expression profile of IFN-u or IFN- ϵ -R1 in zebrafish or xenopus embryos or bird eggs may be informative here.
8. Why did deletion of IFN-u alone result in increased susceptibility to GCRV infection? Please explain why other types of IFN did not provide antiviral protection? Are these IFN systems not redundant in zebra fish?
9. In this respect, it should be shown whether removal of IFN- ϵ -R1 affects ISG induction by IFN- ϕ 1, and IFN- ϕ 3, IFN- γ rel and IFN- γ (as shown in Fig. S2 for other receptors).
10. The authors should perform and show phylogenetic alignments of all sub-members of type I, type III IFN families and IFN-u for one or more species as separate alignments. This should be done for several distinct species, which have all three IFN families represented. Particularly, it should be done for zebra fish (all four type I-like IFNs and IFN-u), clawed frog, chicken, a reptile and a mammalian species (platypus) – separately for each species. This might help to support classification of this new type of IFN.
11. Similarly, alignment of the extracellular domains (ECDs) of the receptors for these IFNs in a given species and sequence comparison of several different species would provide additional information for the classification of the IFN-u antiviral system. It would also be useful to analyze potential STAT recruitment sites on the intracellular domains (ICDs) of these receptors.
12. Because IFN- ϵ binds and signals through a unique class-2 cytokine receptor, the authors should propose to assign a “type IV IFN” designation to the IFN- ϵ genes and proteins.

Reviewer #2 (Remarks to the Author):

The manuscript "IFN- α and its class II cytokine receptors IFN- α R1 and IL10R2" by Shan Nan Chen , Zhen Gan , Jing Hou , Yue Cong Yang , Lin Huang , Bei Huang , Su Wang , and Pin Nie, reports the identification of a new IFN gene in zebrafish that appears to have orthologs in many vertebrates but not in all mammals.

This is potentially a very important discovery that will foster new research in antiviral biology.

General comments:

While this work and other previous works of similar quality allow the unambiguous identification of receptor components for fish class II helical cytokines, a similar high-quality investigation still lacks for the identification of IL10R2 that is based only on over-expression experiments without confirmation based on knockdown or knockout experiments. For this reason, I consider that the authors should not use the IL10R2 name for CRFB4, but just mention the possibility and insist on the fact that this still has to be confirmed.

The authors report knocking down 13 different CRFBs and knocking out CRFB4 and CRFB12. As other authors have reported developmental defects in similar knocked down or knocked out fish, they should report the analysis of putative such defects in their experiments and be very clear on whether or not their IFN α -/-, CRFB4/- and CRFB12/- fish may grow to adulthood and be fertile.

It is not acceptable that an article reporting the identification of a new gene would not include the nucleotide sequence and/or the GenBank accession number. It is not possible to review the work if we do not have access to the nucleotide sequence of the identified gene.

Mandatory revisions:

Lines 128-131: The authors shall summarize their strategy in the result section to highlight its critical points. It is necessary that the GenBank accession number be indicated in the article.

Line 167 : "expression of ISGs induced by IFN- α " the authors should stress that it is an "overexpression of IFN- α ".

The authors should be a little more informative on the schedule of the experiment. It is not that easy to find that the expression of the ISGs was tested at 48hpf. As it has been described that the inactivation of some CRFBs leads to developmental problems, the authors should report their observations on these effects of CRFB1 to CRFB17 knockdown.

Lines 174-176: Why have the authors decided to perform the RT-QPCR analysis at 48hpf after micro-infection? Please explain your choice.

Since it has been reported that knocking down some of the CRFBs may lead to severe developmental problems in the zebrafish, it is necessary that the authors report their observations on how development is impaired by knocking down CRFB1 to CRFB17.

Same request for the CRFB4 and CRFB12 knockout fish.

Line 301: "...type III IFNs from human genomic sequence data (50,51). The cited references are ambiguous because this suggests that the three steps (first, secondly, thirdly) have been described by references 50 & 51, while it is mainly the first step that is similar to what is described in these references.

Please change.

Fig1:

Since introns and their phase have been instrumental in the pipeline that led to the discovery of IFN α , intron positions should be indicated along the protein sequence on fig1a, and intron phases should be indicated on fig 1B and on Sup Fig1.

Fig2, Fig3, Fig4, FigS1B, FigS2A, FigS8B-H, :

When depicting values that vary a lot from sample to sample, the authors should use log scales.

FigS2 C to I:

Please indicate the signification of the dashed lines. Have they been verified experimentally and how ?

FigS4:

Please indicate the positions of the introns and their phases.

FigS5:

Please indicate the positions of the introns and their phases.

Table S1:

All references are missing.

What does "replication" mean? Would "replication" stand for "duplication"?

Minor:

Line 408 : Though ?

Point-to-point responses to reviewers' comments

Reviewer #1 (Remarks to the Author):

1. On page 5 under “Sequence characteristics of IFN- ν ”, the authors state that they deposited the full-length cDNA sequence for the zebrafish *Ifnv* gene in GenBank, but I cannot find this sequence in GenBank. Please provide the GenBank accession number for the sequence that you deposited.

Response: Sorry for the non-accessibility, due to the unpublished data constraints in the database. Now, the sequences of clawed frog and zebrafish *ifnu* genes can be found in GenBank with the accession numbers: MW924834 (<https://www.ncbi.nlm.nih.gov/nuccore/MW924834.1/>), and MW547062 (<https://www.ncbi.nlm.nih.gov/nuccore/MW547062.1/>), respectively, which are now available online.

2. The authors should also deposit and provide the amino acid sequence for their zebrafish IFN- ν protein in GenBank and provide the corresponding accession number.

Response: Yes, thanks. Protein sequences of clawed frog and zebrafish *ifnu* have been deposited in GenBank with the accession numbers, UBY00467.1 (<https://www.ncbi.nlm.nih.gov/protein/2106134352>), and UBY00466.1 (<https://www.ncbi.nlm.nih.gov/protein/2106134350>), respectively.

3. The Discussion section should include some text to explain why there is no IFN- ν gene in placental mammals.

Response: Thank you. A paragraph has been added in the Discussion section to discuss the loss of IFNU gene in placental mammals (Line 353-371):

It is certainly interesting to discuss the evolutionary presence of IFN- ν gene in the Theria, including marsupials and placentals. Despite the presence of IFN- ν in highly conserved ADARB2-PFKP locus in vertebrates as discussed above, IFN- ν is missing in human genome, and cannot be found on the collinear locus of other mammals, even with relatively good quality assembly of their genomes, in the Metatheria and Eutheria, such as Tasmanian devil (*Sarcophilus harrisii*), yellow-footed antechinus (*Antechinus flavipes*), gray short-tailed opossum (*Monodelphis domestica*), common brushtail (*Trichosurus vulpecula*), monito del monte (*Dromiciops gliroides*), Cape golden mole (*Chrysochloris asiatica*), pig (*Sus scrofa*), Arabian camel (*Camelus dromedarius*), African savanna elephant (*Loxodonta africana*) and mouse (*Mus musculus*). Interestingly, IFN- ν R1 gene, as IFN- ν , is also found only in species from the Monotremata (egg-laying mammals), such as platypus, but not in the Theria. In fact, the gene arrangement of CLIC2-FAAH2 locus, where IFN- ν R1 is located in platypus, is inverted in Theria species when compared with

platypus. It is thus considered that chromosome rearrangement might have resulted in the gene loss of IFN- ν R1 in Theria species under the selective pressure, as the gene loss of a receptor or a ligand may cause specifically the loss of the ligand-receptor system in evolution⁵⁹. However, this may provide an indirect evidence to support the ligand-receptor system of IFN- ν and IFN- ν R1.

Reference

59. Grandchamp A, Monget P. Synchronous birth is a dominant pattern in receptor-ligand evolution. *BMC Genomics* **19**, 611 (2018).

4. The implication of transition from unique receptor chains for type I and type II IFNs to shared receptors for type III IFNs and other class-2 cytokines is interesting, especially because IFN- ν retains the “type I IFN” C-terminus. The authors should discuss why IFN- ν might represent a missing link between the type-I and type-III IFN gene families.

Response: Thank you for raising this point. We have added a paragraph, as shown below, in the Discussion section to discuss the issue (Line 372-393):

It has been reported that class II cytokines and class II cytokine receptors might have undergone duplication to form specialized distinct ligand-receptor systems⁶⁰, and this may represent an evolutionary foundation for functional diversification of genes⁶¹. In this study, IFN- ν gene shows the similar C-terminal sequence as type I IFNs, and shares the signal-transduction receptor, IL10R2 as type III IFNs, which is also shared among some IL-10 family cytokines². Thus, it may be implied that IFN- ν and type I/III IFNs share a common ancestor. In cartilaginous fish, the coexistence of type I, III IFN and IFN- ν genes is observed, indicating that these three types of IFNs might have diverged before the appearance of cartilaginous fish, and IFN- ν gene and type I/III IFNs may have undergone independent evolution to form specific ligand-receptor pairs, including IFN- ν / IFN- ν R1, type I IFN / IFNAR2, and type III IFN / IFNLR1. Therefore, it can be hypothesized that the loss of IFN- ν gene in the Theria might be resulted from the lack of IFN- ν R1 due to the chromosome rearrangement, which may, on the other hand, provide indirect evidence to support the ligand-receptor system of IFN- ν and IFN- ν R1. In addition, intronless type I IFNs in amniotes might have originated from retroposition of intron-containing type I IFN transcripts in lower vertebrates^{17,19,62}, Mammalian intronless type I IFNs may have undergone further expansion to form a few more subtypes, but some of them do not exist in all mammal lineages, such as IFN- ϵ and IFN- τ , which are found in placentals but not in marsupials⁶³. IFN- ν gene might have been lost possibly before the divergence of placentals, and it seems likely that IFN- ν gene and type I IFN subtypes may have undergone independent evolution in mammals

References

2. Ouyang WJ, Rutz S, Crellin NK, Valdez PA, Hymowitz SG. Regulation and functions of the IL-10 family

- of cytokines in inflammation and disease. *Annu. Rev. Immunol.* **29**, 71-109 (2011).
17. Gan Z, et al. Unique composition of intronless and intron-containing type I IFNs in the Tibetan frog *Nanorana parkeri* provides new evidence to support independent retroposition hypothesis for type I IFN genes in amphibians. *J. Immunol.* **201**, 3329-3342 (2018).
19. Qi ZT, Nie P, Secombes CJ, Zou J. Intron-containing type I and type III IFN coexist in amphibians: refuting the concept that a retroposition event gave rise to type I IFNs. *J. Immunol.* **184**, 5038-5046 (2010).
60. Krause CD, Pestka S. Evolution of the Class 2 cytokines and receptors, and discovery of new friends and relatives. *Pharmacol. Ther.* **106**, 299-346 (2005).
61. Mandrioli M, Malagoli D, Ottaviani E. Evolution game: which came first, the receptor or the ligand? *ISJ-Invertebr. Surviv. J.* **4**, 51-54 (2007).
62. Chen SN, Gan Z, Nie P. Retroposition of the long transcript from multiexon IFN-beta homologs in ancestry vertebrate gave rise to the proximal transcription elements of intronless IFN-beta promoter in humans. *J. Immunol.*, <https://doi.org/10.4049/jimmunol.2100092> (2021).
63. Krause CD, Pestka S. Cut, copy, move, delete: The study of human interferon genes reveal multiple mechanisms underlying their evolution in amniotes. *Cytokine* **76**, 480-495 (2015).

5. The sequence alignments shown in Fig. S4 did not include **a broad range** of species, and for the ligands, were not structurally based. A structure-based alignment of **several** type I IFNs, type II IFNs, type III IFNs and IFN- ν should be performed at the same time and shown in the same table.

Response: Thanks. Additional sequence alignments of IFNU genes from over eighty species, including fish, amphibians, reptiles, birds and mammals (Australian echidna, *Tachyglossus aculeatus*), respectively, have been shown in the current Supplementary Fig. 12.

We also collected IFNU, type I, II, and III IFN sequences from the five representing species, including zebrafish (IFN1 [PDB: 3piv], IFN2 [PDB: 3piw], IFNG and IFNU), clawed frog (IFN1, IFNL, IFNG and IFNU), green anole (IFN1, IFNL, IFNG and IFNU), chicken (IFNB, IFNL, IFNG and IFNU) and platypus (IFN1, IFNL3L, IFNG and IFNU) to perform the prediction of protein structures by using AlphaFold 2. Structural comparison showed that the integral structure of these IFN- ν proteins from zebrafish to platypus resembles more their own type I IFNs with root mean square deviation (RMSD) value being 4.167-7.633 Å (across all pruned atom pairs) than type II (RMSD, 19.830-29.279 Å) or III IFNs (RMSD, 8.701-11.533 Å), respectively (Supplementary Figs. 18-22).

In addition to the similar structure at C-terminal region, the N-terminal region between type I IFN and IFN- ν could not be well overlaid, with the corresponding Helix A region representing the noticeable difference (Supplementary Figs. 18-22). It is thus indicated that IFN- ν gene is obviously distinct from type I, II and III IFNs in consideration of sequence and putative structure.

6. Similar to my comment regarding Fig. S4, the sequence alignment of IFN- ϵ -R1 proteins in Fig. S5 should include an alignment of other class-2 receptor proteins, including IFN- α R1, IFN- α R2, IFN- λ R1 and IL-10R2.

Response: Thanks. Additional protein sequences, including IFNUR1, IFNAR1, IFNAR2, IFNLR1 and IL10R2 genes, from fish to mammal were collected to perform re-alignment and shown in the current Supplementary Fig. 23. IFN- ν R1 genes were also identified in vertebrates from fish to mammal, and the intron phases of most IFN- ν R1 genes are conserved, being 1-2-1-0-1-0, when compared with receptor genes of other IFNs (Supplementary Fig. 23). IFN- ν R1 consists of conserved structures of class II cytokine receptors, such as signal peptide, extracellular (with a D200 domain containing two FNIII-like domains), transmembrane and intracellular regions (Supplementary Fig. 23).

7. There is no discussion as to why IFN- ϵ s are lost in placental mammalian genomes? The correlation between placental structure (and IFN- ϵ and IFN- τ being placental IFNs) and the loss of IFN- ν needs some discussion here. Are they involved in establishing antiviral immunity in the laid egg? Are they expressed most strongly in embryos? An in situ expression profile of IFN- ν or IFN- ν R1 in zebrafish or xenopus embryos or bird eggs may be informative here.

Response: Thanks. The discussion about the reason why IFN- ϵ s are lost in placental mammalian genomes has been added in the Discussion section (Line 353-371). We have also conducted whole-mount in situ hybridization and also ISG examination in zebrafish embryos (Supplementary Fig. 6) as suggested.

As discussed in the revision (Line 353-371), intronless type I IFNs in amniote might have originated from retroposition of intron-containing type I IFN transcripts in lower vertebrates (Qi et al., 2010; Gan et al., 2017, 2018). Mammalian intronless type I IFNs have a further gene expansion to form a few subtypes, while some of them do not exist all mammal lineage, such as IFN- ϵ and IFN- τ , which are found in placentals but not in marsupials (Krause and Pestka 2015). In fact, IFN- ν gene had been lost possibly before the divergence of placentals, and it is considered that IFN- ν gene and type I IFN subtypes may have undergone independent evolution.

Whole-mount in situ hybridization (WISH) and qRT-PCR was performed to detect the expression of IFN- ν R1 gene in zebrafish embryos (24 hpf). As shown in Supplementary Fig. 6, IFN- ν R1 gene has wide expression in zebrafish embryos. In addition, the expression of the ISGs induced by IFN- ν overexpression was also detected at 24 hpf (Supplementary Fig. 1). It was observed that IFN- ν has a strong induction of ISGs at 24 hpf. In fact, effects of CRFB knockdown on the expression of the ISGs induced by IFN- ν overexpression was also detected at 24 hpf, and the results are similar to those at 48 hpf and were shown in Supplementary Fig. 5, which also indicates that both CRFB4/IL10R2 and CRFB12/ IFN- ν R1 are associated with IFN- ν -mediated signalling. It is suggested that IFN- ν signalling is involved in establishing antiviral immunity in zebrafish embryos.

References:

Gan Z. et al. Intronless and intron-containing type I IFN genes coexist in amphibian *Xenopus tropicalis*: Insights into the origin and evolution of type I IFNs in vertebrates. *Dev. Comp. Immunol.* 67, 166-176 (2017).

Gan Z. et al. Unique composition of intronless and intron-containing type I IFNs in the Tibetan frog *Nanorana parkeri* provides new evidence to support independent retroposition hypothesis for type I IFN genes in amphibians. *J. Immunol.* 201, 3329-3342 (2018).

Qi, Z. et al. Intron-containing type I and type III IFN coexist in amphibians: refuting the concept that a retroposition event gave rise to type I IFNs. *J. Immunol.* 184, 5038-5046 (2010).

Krause C.D. & Pestka S. Cut, copy, move, delete: The study of human interferon genes reveal multiple mechanisms underlying their evolution in amniotes. *Cytokine* 76 480–495 (2015).

8. Why did deletion of IFN- ν alone result in increased susceptibility to GCRV infection? Please explain why other types of IFN did not provide antiviral protection? Are these IFN systems not redundant in zebra fish?

Response: Thanks. In this study, antiviral signalling induced by IFN- ν has been proved to be associated with the specific receptor (IFN- ν R1/CRFB12) and the “common receptor” (CRFB4/IL10R2), which is not used by type I or type II IFN. Thus, we considered that the independent receptor systems for IFN- ν is a potential reason to explain why the deletion of IFN- ν alone results in increased susceptibility to GCRV infection, which indicates that IFN- ν contributes to host antiviral activity. In fact, other types of IFNs are also working in antiviral response. As shown in the current Supplementary Fig. 3, knockdown of type I (IFN1) and type II IFN (IFNG and IFNGrel) further increases the susceptibility to GCRV infection in *ifnu* deficiency fish. Therefore, it is suggested that type I, type II IFN and IFN- ν gene are not redundant in antiviral systems of zebrafish.

9. In this respect, it should be shown whether removal of IFN- ν R1 affects ISG induction by IFN- ϕ 1, and IFN- ϕ 3, IFN- γ rel and IFN- γ (as shown in Fig. S2 for other receptors).

Response: Thanks. As shown in the current Supplementary Fig. 4, the removal of IFN- ν R1 has no effects on ISG induction by type I (IFN- ϕ 1 and IFN- ϕ 3) and type II (IFN- γ rel and IFN- γ) IFN.

10. The authors should perform and show phylogenetic alignments of all sub-members of type I, type III IFN families and IFN- ν for one or more species as separate alignments. This should be done for several distinct species, which have all three IFN families represented. Particularly, it should be done for zebra fish (all four

type I-like IFNs and IFN- ν), clawed frog, chicken, a reptile and a mammalian species (platypus) – separately for each species. This might help to support classification of this new type of IFN.

Response: Thanks for the suggestion, and this is done in the revision.

We collected all possible sub-members of type I, type II, type III IFNs and IFN- ν from zebrafish (with the absence of type III IFN), clawed frog, green anole, chicken and platypus from NCBI database, respectively. Separate alignments of IFN- ν and the members from other three IFN types were performed to reveal the sequence differences, IFN- ν s have low protein sequence identity (5.5-17.1%, 2.6-13.6%, 7.4-15.5%, 11.1-15.1% and 4.8-19.0%) with all the IFN subfamilies in zebrafish, clawed frog, green anole, chicken and platypus, respectively (Supplementary Figs. 13-17). Separate phylogenetic analysis showed that IFN- ν gene is in a separate clade from all the three types IFN (Supplementary Figs. 13-17).

11. Similarly, alignment of the extracellular domains (ECDs) of the receptors for these IFNs in a given species and sequence comparison of several different species would provide additional information for the classification of the IFN- ν antiviral system. It would also be useful to analyze potential STAT recruitment sites on the intracellular domains (ICDs) of these receptors.

Response: Thanks, and this is done in the revision. The alignments of the extracellular domains (ECDs) and the intracellular domains (ICDs) of the IFN receptors were also performed. Most sequences related to the beta-sheet structures in IFN- ν R1 extracellular region are conserved when compared to other IFN receptors from fish to mammal (Supplementary Fig. 24).

On the other hand, alignments showed that membrane-proximal region of vertebrate IFN- ν R1 intracellular sequences possesses a potential docking site for JAK family, which contains a conserved ‘box1’ membrane-proximal receptor peptide motif with PXXL sequence and a hydrophobic residues-rich ‘box2’ receptor motif (Supplementary Fig. 25). In fact, the ‘box1’ and ‘box2’ sequences of class II cytokine receptors have been proved as the critical motifs for binding of JAK family members (Morris et al., 2018). Moreover, two highly conserved tyrosines were found in IFN- ν R1 intracellular regions from fish to mammal, which are believed to be associated with STAT activation, and will be verified experimentally in future (Supplementary Fig. 25).

References:

Morris R., Kershaw, N.J., Babon J.J. The molecular details of cytokine signaling via the JAK/STAT pathway. *Protein Science* 2018(27): 1984-2009.

12. Because IFN- ν binds and signals through a unique class-2 cytokine receptor, the authors should propose to assign a “type IV IFN” designation to the IFN- ν genes and proteins.

Response: Thank you very much for such a detailed review and the valuable comments, which is greatly appreciated.

Reviewer #2 (Remarks to the Author):

The manuscript “IFN- ν and its class II cytokine receptors IFN- ν R1 and IL10R2” by Shan Nan Chen , Zhen Gan , Jing Hou , Yue Cong Yang , Lin Huang , Bei Huang , Su Wang , and Pin Nie, reports the identification of a new IFN gene in zebrafish that appears to have orthologs in many vertebrates but not in all mammals.

This is potentially a very important discovery that will foster new research in antiviral biology.

Response: Thank you very much for your careful reading and for your positive comments, which is greatly appreciated.

General comments:

While this work and other previous works of similar quality allow the unambiguous identification of receptor components for fish classII helical cytokines, a similar high-quality investigation still lacks for the identification of IL10R2 that is based only on over-expression experiments without confirmation based on knockdown or knockout experiments. For this reason, I consider that the authors should not use the IL10R2 name for CRFB4, but just mention the possibility and insist on the fact that this still has to be confirmed.

Response: Thank you very much for pointing out this. We fully agree with the view due to the insufficient investigation for identification of *crfb4* in fish. The name of zebrafish IL10R2 has been changed to *crfb4* in the manuscript, including the title.

The authors report knocking down 13 different CRFBs and knocking out CRFB4 and CRFB12. As other authors have reported developmental defects in similar knocked down or knocked out fish, they should report the analysis of putative such defects in their experiments and be very clear on whether or not their IFN ν ^{-/-}, CRFB4^{-/-} and CRFB12^{-/-} fish may grow to adulthood and be fertile.

Response: Thank you. The knockdown of *crfb4* or *crfb5* by morpholino is lethal to zebrafish embryos, and severe developmental defects were also found in *crfb4* and *crfb5* knockdown zebrafish which become dead within 3 dpf (Supplementary Figs. 10-11). Knockdown of other CRFBs has no such effects on embryo development (Supplementary Figs. 10-11). Interestingly, severe developmental defects were not found in *crfb4* knockout zebrafish, and not in *crfb12* and *ifnu* deficiency fish (Supplementary Figs. 8-9). It is also found that all these gene deficiency fish, including *crfb4*^{-/-}, *crfb12*^{-/-} and *ifnu*^{-/-}, can grow to adulthood and reproduce normally. We can provide videos upon request, but we only include the spawning video for *crfb4* knockout 4-month old zebrafish as Supplementary Video 1.

In our research, we did not initially have a plan to develop knockout for zebrafish *crfb4* gene due to the lethality of *crfb4* knockdown. However, a huge project

regarding genome editing of zebrafish genes has been launched and carried out by China Zebrafish Resource Center (CZRC, the leader: Prof. Yonghua Sun), and partial results have been published (Genome editing of zebrafish Chromosome 1, Sun et al., 2020). It is clear that *crfb4* knockout plan is included in the project, and a 5 bp deletion (CAGTG) of *crfb4* mutant zebrafish (<http://zfin.org/ZDB-ALT-191230-6>) was generated by Dr. Xunwei Xie (Technical Staff) in the CZRC. Surprisingly, *crfb4* knockout does not lead to lethality.

It has been reported that the knockout and knockdown of some genes show phenotypic differences in model systems including zebrafish, Arabidopsis, and mouse (Sztal and Stainier, 2020). Although knockdown and knockout of *crfb4* reveal inconsistent and different phenotypes regarding embryonic development in zebrafish, the removal of CRFB4 resulted from both knockdown and knockout all impaired antiviral ISG expression induced by IFN- ν . Meanwhile, IFN- ν signalling is abolished in *crfb12* deficiency zebrafish. Consequently, these data support the key conclusion that CRFB12 and CRFB4 are involved in IFN- ν -mediated signalling.

References:

Sun, Y., Zhang, B., Luo, L., Shi, D.L., Wang, H., Cui, Z., Huang, H., Cao, Y., Shu, X., Zhang, W., et al. (2020). Systematic genome editing of the genes on zebrafish Chromosome 1 by CRISPR/Cas9. *Genome Research* 30, 118-126.

Sztal, T.E., Stainier, D.Y.R. (2020). Transcriptional adaptation: a mechanism underlying genetic robustness. *Development* 147, dev186452.

It is not acceptable that an article reporting the identification of a new gene would not include the nucleotide sequence and/or the GenBank accession number. It is not possible to review the work if we do not have access to the nucleotide sequence of the identified gene.

Response: Sorry about this. They are now fully accessible. The sequences of clawed frog and zebrafish ifnu genes have been deposited in GenBank, with the accession numbers of clawed frog and zebrafish ifnu genes are MW924834 (<https://www.ncbi.nlm.nih.gov/nuccore/MW924834.1/>) and MW547062 (<https://www.ncbi.nlm.nih.gov/nuccore/MW547062.1/>), respectively, which are now available online.

Mandatory revisions:

Lines 128-131: The authors shall summarize their strategy in the result section to highlight its critical points. It is necessary that the GenBank accession number be indicated in the article.

Response: Thanks, the strategy with the critical points was added to the result section, Line 124-131:

Since some fish genes belonging to class II cytokines cannot be well annotated due to their low expression levels, bioinformatic strategies (see Methods) were employed to scan available genomic sequences, and unannotated sequences in intergenic regions

between annotated genes in zebrafish genome (assembly version: Zv9) were collected to carry out gene prediction and/or annotation on the basis of class II cytokine gene features, including 1) five-coding-exon organization, 2) intron phase being zero, 3) signal peptide present at N-terminal region, and 4) putative protein sequence containing multiple alpha helices.

The GenBank accession number of clawed frog and zebrafish ifnu gene is MW924834 and MW547062, respectively, which are listed into the manuscript (Line 277 and Line 134).

Line 167 : “expression of ISGs induced by IFN- ν ” the authors should stress that it is an “overexpression of IFN- ν ”.

The authors should be a little more informative on the schedule of the experiment. It is not that easy to find that the expression of the ISGs was tested at 48hpf. As it has been described that the inactivation of some CRFBs leads to developmental problems, the authors should report their observations on these effects of CRFB1 to CRFB17 knockdown.

Response: Thanks, “overexpression of IFN- ν ” has been added up (Line 174).

The knockout and knockdown effect has been asked by another reviewer, and we have shown the detailed phenotype observations, which can then provide fine replies to this concern. In particular, we found that the knockdown of *crfb4* or *crfb5* is lethal to zebrafish embryo as previously reported, and severe developmental defects were also observed in *crfb4* and *crfb5* knockdown zebrafish embryos which are dead within 3 dpf (Supplementary Figs. 10-11). The knockdown of other CRFBs has no such effects on embryo development (Supplementary Figs. 10-11). However, at 48 hpf, several embryos of *crfb4* and *crfb5* knockdown are still alive but are dying, and these dying embryos were collected for further experiments.

Lines 174-176: Why have the authors decided to perform the RT-QPCR analysis at 48hpf after micro-infection? Please explain your choice.

Since it has been reported that knocking down some of the CRFBs may lead to severe developmental problems in the zebrafish, it is necessary that the authors report their observations on how development is impaired by knocking down CRFB1 to CRFB17. Same request for the CRRFB4 and CRFB12 knockout fish.

Response: Thanks. In the revision, the expression of ISGs induced by IFN- ν overexpression was detected at 24, 48 and 72 hpf, separately (Supplementary Fig. 1). It was observed that IFN- ν has stronger induction of ISGs at 48 hpf, and thus we considered to detect the expression of the ISGs at 48 hpf. In the revision, effects of CRFB knockdown on the expression of the ISGs induced by IFN- ν overexpression were also detected at 24 hpf, and the results are similar to that at 48 hpf and were shown in Supplementary Fig. 5, which also indicates that both CRFB4 and CRFB12 are associated with IFN- ν -mediated signalling.

The knockdown of *crfb4* or *crfb5* is lethal to zebrafish embryo, and severe developmental defects were also found in *crfb4* and *crfb5* knockdown zebrafish (Supplementary Figs. 10-11). The knockdown of other CRFBs has no such effects on embryo development (Supplementary Figs. 10-11). In addition, severe developmental defects were not found in *crfb4* knockout zebrafish, and not in *crfb12* and *ifnu* deficiency fish (Supplementary Figs. 8-9). It is also found that all these gene deficiency fish, including *crfb4*^{-/-}, *crfb12*^{-/-} and *ifnu*^{-/-}, can grow to adulthood and reproduce normally (Supplementary Figs. 8-9). We have only submitted *crfb4*^{-/-} video as Supplementary Video 1, and can provide other videos on *crfb12*^{-/-} and *ifnu*^{-/-} upon request.

Line 301: "...type III IFNs from human genomic sequence data (50,51). The cited references are ambiguous because this suggests that the three steps (first, secondly, thirdly) have been described by references 50 & 51, while it is mainly the first step that is similar to what is described in these references.

Please change.

Response: Thanks for pointing out this. The inappropriate citation has been changed (Line 405-407).

The following suggestions are quite straight forward, and have been fully accepted in the revision.

Fig1:

Since introns and their phase have been instrumental in the pipeline that led to the discovery of IFNU, intron positions should be indicated along the protein sequence on fig1a, and intron phases should be indicated on fig 1B and on Sup Fig1.

Response: Thanks. The intron phases of IFNU have been indicated in Fig. 1a, Fig. 1b and the current Supplementary Fig. 2.

Fig2, Fig3, Fig4, FigS1B, FigS2A, FigS8B-H, :

When depicting values that vary a lot from sample to sample, the authors should use log scales.

Response: Thanks for the suggestion. The log scales have been applied to all related figures, including Fig. 2, Fig. 3, Fig. 4, Supplementary Fig. 1, Supplementary Fig. 2, Supplementary Fig. 5a-b and Supplementary Fig. 29.

FigS2 C to I:

Please indicate the signification of the dashed lines. Have they been verified experimentally and how ?

Response: Thanks. The dashed lines indicate transcript structure of CRFB genes around the block site by morpholino after knockdown. The mutant transcripts after

amplification have been sequenced and the results for sequencing were shown in Supplementary Fig. 4.

FigS4:

Please indicate the positions of the introns and their phases.

Response: Thanks. The positions of the introns and their phases have been indicated in Supplementary Fig. 12. Intron phases of all the detected IFNU genes are zero.

FigS5:

Please indicate the positions of the introns and their phases.

Response: Thanks, and the positions of the introns and their phases have been indicated in Supplementary Fig. 23. IFN- ν R1 genes were identified in many vertebrates from fish to mammal, and the intron phases of most IFN- ν R1 genes are conserved 1-2-1-0-1-0 compared with other IFN receptor genes (Supplementary Fig. 23).

Table S1:

All references are missing.

What does “replication” mean? Would “replication” stand for “duplication”?

Response: Thanks, and references have been added, and “replication” has been changed to “duplication”.

Minor:

Line 408 : Though ?

Response: Thanks, the sentence has been rewritten.

REVIEWER COMMENTS

Reviewer #1 (Remarks to the Author):

The authors have provided additional data as requested. The revised manuscript is much improved. I just have one minor change request:

The term, CRFB4, is outdated and should be replaced by "IL10RB" when referring to the gene and "IL-10R2" when referring to the corresponding protein. The authors should replace CRFB4 with IL10RB and IL-10R2 throughout the text.

Reviewer #2 (Remarks to the Author):

In the revised version of their manuscript "IFN- α and its class II cytokine receptors IFN- α 1 and IL10R2" by Shan Nan Chen, Zhen Gan, Jing Hou, Yue Cong Yang, Lin Huang, Bei Huang, Su Wang, and Pin Nie, have very well addressed most of the issues raised by the reviewers.

As I pointed out in my first review, this is a very important discovery that will foster new research in antiviral biology.

I have very much appreciated, not only their responses to my comments, but especially their use of AlphaFold2 to address the request of reviewer1 related to structural alignments.

I only have a few requests that may be dealt directly by the editor ... but I am open to be included in the extra review process of these changes. While the requests on the text are minor, I consider that their annotations of log scales are not correct and should be changed to more self-understandable representations, i.e. a format that their future readers are used to.

Line 143 "organization of type I IFNs in amniotes" should be replaced by "organization of the intronless type I IFN genes in amniotes".

Lines 175 & 176 "when all class II cytokine receptors, except il22bp (CRFB9) and the duplicated tissue factor genes (CRFB10 and 11),"

Please use uppercase for the first letters of "Tissue Factor"

Line 235 "intron phases of most IFN- α 1 genes are conserved, being 1-2-1-0-1-0," I suggest that the authors include a reference as for example, their reference 9.

Lines 281 & 282: "Recombinant IFN- α protein", do you mean "Recombinant clawed frog IFN- α protein" ?

Fig1b, Sup2a and Sup29e: Since the phase is that of an intron, not that of a 5' splice site, I suggest that the "0" should be placed in the middle of the intron, not next to the 5' site. (See example included.)

Figures: "decimal logarithms" not "denary". Notwithstanding the decimal vs denary, what you display are not be "denary logarithm", but decimal values represented on a \log_{10} scale. (See an included example.)

For all figures with viral titers that are so high, please start the scale just below the difference between your different samples. (see example included)

Fig2e and Sup6b: From a theoretical point of view, using an increment of 0.5 on a log scale should not be a problem. However, few readers of your article are aware of the fact that $\log_{10}(0.0316)=-1$. I therefore think that it is really not a good idea to have an increment of 0.5 on your ordinate scale. Please stick to the increment of 1, and use the labelling 10^2 and 10^{-1}

REVIEWERS' COMMENTS:

Reviewer #1 (Remarks to the Author):

The authors have provided additional data as requested. The revised manuscript is much improved. I just have one minor change request:

The term, CRFB4, is outdated and should be replaced by "IL10RB" when referring to the gene and "IL-10R2" when referring to the corresponding protein. The authors should replace CRFB4 with IL10RB and IL-10R2 throughout the text.

Response: Thank you very much for the positive comment and valuable suggestion. The CRFB4 have been replaced by IL10RB and IL-10R2 throughout the text, except in the necessary descriptions concerning annotation.

Reviewer #2 (Remarks to the Author):

In the revised version of their manuscript “IFN- γ and its class II cytokine receptors IFN- γ R1 and IL10R2” by Shan Nan Chen, Zhen Gan, Jing Hou, Yue Cong Yang, Lin Huang, Bei Huang, Su Wang, and Pin Nie, have very well addressed most of the issues raised by the reviewers.

Response: Thank you very much for your positive comment.

As I pointed out in my first review, this is a very important discovery that will foster new research in antiviral biology.

I have very much appreciated, not only their responses to my comments, but especially their use of AlphaFold2 to address the request of reviewer1 related to structural alignments.

Response: We are grateful for your appreciation.

I only have a few requests that may be dealt directly by the editor ... but I am open to be included in the extra review process of these changes. While the requests on the text are minor, I consider that their annotations of log scales are not correct and should be changed to more self-understandable representations, i.e. a format that their future readers are used to.

Response: Thank you for the valuable and helpful suggestions. We have changed all the scales as suggested.

Line 143 “organization of type I IFNs in amniotes” should be replaced by “organization of the intronless type I IFN genes in amniotes”.

Response: Thanks, changed.

Lines 175 & 176 “when all class II cytokine receptors, except il22bp (CRFB9) and the duplicated tissue factor genes (CRFB10 and 11),”

Please use uppercase for the first letters of “Tissue Factor”

Response: Thanks, done.

Line 235 “intron phases of most IFN- ν R1 genes are conserved, being 1-2-1-0-1-0,” I suggest that the authors include a reference as for example, their reference 9.

Response: Thanks, the reference has been cited.

Lines 281 & 282: “Recombinant IFN- ν protein”, do you mean “Recombinant cloned frog IFN- ν protein” ?

Response: Yes, thanks.

Fig1b, Sup2a and Sup29e: Since the phase is that of an intron, not that of a 5' splice site, I suggest that the “0” should be placed in the middle of the intron, not next to the 5' site. (See example included.)

Response: Thanks, and the number (“0”) has been placed in the middle.

Figures: “decimal logarithms” not “denary”. Notwithstanding the decimal vs denary, what you display are not be “denary logarithm”, but decimal values represented on a \log_{10} scale. (See an included example.)

Response: Thanks, it has been changed.

For all figures with viral titers that are so high, please start the scale just below the difference between your different samples. (see example included)

Response: Thanks, the scale changed as suggested.

Fig2e and Sup6b: From a theoretical point of view, using an increment of 0.5 on a log scale should not be a problem. However, few readers of your article are aware of the fact that $\log_{10}(0.0316)=-1$. I therefore think that it is really not a good idea to have an increment of 0.5 on your ordinate scale. Please stick to the increment of 1, and use the labelling 10^{-2} and 10^{-1}

Response: Thanks, it has been changed.